# MARTI: A Framework for Multi-Agent LLM Systems Reinforced Training and Inference

**Kaiyan Zhang**[1,8†*]  **Kai Tian**[1,8*]  **Runze Liu**[1,6*]  **Sihang Zeng**[3]  **Xuekai Zhu**[2]  **Guoli Jia**[1]
**Yuchen Fan**[2,6]  **Xingtai Lv**[1]  **Yuxin Zuo**[1]  **Che Jiang**[1]  **Yuru Wang**[1]  **Jianyu Wang**[4]
**Ermo Hua**[1,6]  **Xinwei Long**[1]  **Junqi Gao**[5]  **Youbang Sun**[1]  **Zhiyuan Ma**[7]  **Ganqu Cui**[6]
**Ning Ding**[1,6]  **Biqing Qi**[6‡]  **Bowen Zhou**[1,6‡]

[1]Tsinghua University  [2]Shanghai Jiao Tong University  [3]University of Washington
[4]Beijing Institute of Technology  [5]Harbin Institute of Technology
[6]Shanghai AI Laboratory  [7]Huazhong University of Science and Technology  [8]Frontis.AI
[†]Project Leader  [*]Core Contributions  [‡]Corresponding Author
zhang-ky22@mails.tsinghua.edu.cn  ⏻ MARTI

## Abstract

We present MARTI (Multi-Agent Reinforced Training and Inference), an open-source framework designed to facilitate scalable and efficient learning of multi-agent LLM systems. MARTI supports centralized multi-agent interactions and distributed policy training, with the added capability of multi-turn asynchronous rollouts to enhance training efficiency. The framework includes dynamic workflows for multi-agent interactions, which integrate both rule-based verifiable rewards and LLM-based generative rewards. We validate the effectiveness of MARTI through comprehensive experiments on diverse mathematical tasks, demonstrating that multi-agent LLM-based systems outperform single-agent systems within the same inference budget after convergence. Our contributions lay the foundation for exploring scalable collaborations within LLM-based multi-agent systems and advancing the capabilities of large reasoning models.

## 1 Introduction

Large Reasoning Models (LRMs), such as DeepSeek-R1 (Guo et al., 2025) and OpenAI o1/o3 (El-Kishky et al., 2025), highlight the significant role Reinforcement Learning (RL) plays in enhancing the reasoning capabilities of Large Language Models (LLMs) for solving complex problems. Notably, LRMs can explore and generate extended chains of thought using only rule-based outcome rewards. This RL paradigm has also demonstrated considerable progress in other domains, including visual reasoning (Liu et al., 2025d; Zhou et al., 2025; Team et al., 2025) and agentic reasoning (Wang et al., 2025c; Jin et al., 2025) tasks. These studies indicate the effectiveness of scaling up test-time inference computations using RL. However, further performance improvements through post-training RL typically demand substantial computational resources. Additionally, recent research suggests that RL primarily activates intrinsic capabilities and reflective patterns established during pre-training (Gandhi et al., 2025; Yue et al., 2025a; Shah et al., 2025). Consequently, the initial model's passk performance sets an upper bound for RL-based enhancements (Yue et al., 2025a), which means the base model determines the reasoning limit. Therefore, the most viable approach for significantly boosting policy model performance remains within the scaling laws (Kaplan et al., 2020; Brown et al., 2020), either by training models on larger datasets or increasing the model's parameter size. Regarding the reinforcement learning stage, effectively leveraging the potential of exploration and environmental interaction remains a critical challenge (Silver & Sutton, 2025).

Meanwhile, LLM-based Multi-Agent Systems (MAS) (Han et al., 2024; Guo et al., 2024) scale inference computation by expanding the number of agents, each adaptively responding to specific tasks. Numerous open-source frameworks for LLM-based MAS are currently available, including AutoGen (Wu et al., 2023a), CAMEL (Li et al., 2023), and MetaGPT (Hong et al., 2024). However, these frameworks predominantly rely on LLM inference. This reliance makes their efficacy highly

Table 1: Comparison between Multi-Agent and RL Framework.

| Framework | MAS Inference | Single RL | MAS RL |
|---|:---:|:---:|:---:|
| CAMEL (Li et al., 2023) | ✓ | ✗ | ✗ |
| AutoGen (Wu et al., 2023b) | ✓ | ✗ | ✗ |
| Meta-GPT (Hong et al., 2023) | ✓ | ✗ | ✗ |
| GPTSwarm (Zhuge et al., 2024) | ✓ | ✗ | ✗ |
| TRL (von Werra et al., 2020) | ✗ | ✓ | ✗ |
| OpenRLHF (Hu et al., 2024a) | ✗ | ✓ | ✗ |
| Verl (Sheng et al., 2024) | ✗ | ✓ | ✗ |
| AReaL (Fu et al., 2025) | ✗ | ✓ | ✗ |
| MARTI (Our) | ✓ | ✓ | ✓ |

dependent on the instruction-following capabilities of the LLMs, a factor that, as recent studies (Pan et al., 2025) indicate, can readily contribute to operational failures. Concurrently, several RL frameworks (e.g., OpenRLHF (Hu et al., 2024b), veRL (Sheng et al., 2025), TRL (von Werra et al., 2020)), designed to train LLMs, can enhance LLM reasoning abilities but do not support LLM-based MAS. This observation prompts an essential question: *can we leverage RL to improve LLM-based MAS and thereby achieve superior reasoning performance?* Addressing this question requires mitigating the gap between inference and RL training in LLM-based MAS, which in turn necessitates a unified framework integrating multi-agent reinforcement learning with inference capabilities.

In this work, we propose the **M**ulti-**A**gent **R**einforced **T**raining and **I**nference (**MARTI**) framework for LLM-based multi-agent systems. MARTI is built upon the OpenRLHF framework (Hu et al., 2024b), which enables scalable and high-performance RL for LLMs. For multi-agent inference, we integrate asynchronous workflows to facilitate dynamic interactions. MARTI employs a centralized interaction design for its built-in workflows (e.g., Multi-Agent Debate, Chain-of-Agents, and Mixture-of-Agents) and customizable workflows, while utilizing distributed policy training for individual agents. During inference, MARTI supports rule-based rewards as used in DeepSeek-R1 (Guo et al., 2025), along with generative reward models (Liu et al., 2025c). Prior to transferring rollout experiences to distributed agent policies, MARTI incorporates several reward shaping techniques (Park et al., 2025; Motwani et al., 2024) and credit assignment strategies to allocate rewards effectively.

Our preliminary experiments demonstrate that MARTI enhances multi-agent workflow performance, achieving a higher upper bound than single-agent RL training under the same inference budget. For instance, our multi-agent debate workflow based on DeepScaleR-1.5B-Preview (Luo et al., 2025) attains a score of 65.0 on the AIME benchmark, surpassing the single-agent baseline (53.5), which relies on large reasoning models with test-time RL. However, challenges remain in multi-agent RL, including the need for improved reward models for multi-agent systems (Pan et al., 2025) and real-world applicability (Li et al., 2025; Zheng et al., 2025). We will continue optimizing MARTI to advance MAS training for high-value applications.

Our contributions can be summarized as follows:

- We propose and open-source the Multi-Agent Reinforced Training Infrastructure (MARTI), a framework that facilitates centralized multi-agent interactions and distributed policy training, enabling scalable multi-agent learning. MARTI also supports multi-turn asynchronous rollouts during training to enhance the efficiency of multi-agent learning.
- We implement dynamic workflows for multi-agent interactions that support both rule-based verifiable rewards and LLM-based generative rewards.
- We conduct comprehensive experiments on various mathematical tasks, which demonstrate that multi-agent LLM-based systems can achieve superior performance than single agent under the same inference budget after convergence.

## 2 MARTI: MULTI-AGENT REINFORCED TRAINING AND INFERENCE

### 2.1 FRAMEWORK DESIGN

We designed the MARTI framework based on the principle of centralized multi-agent interaction with distributed policy training, where all agent interactions and reward allocation occur centrally, while

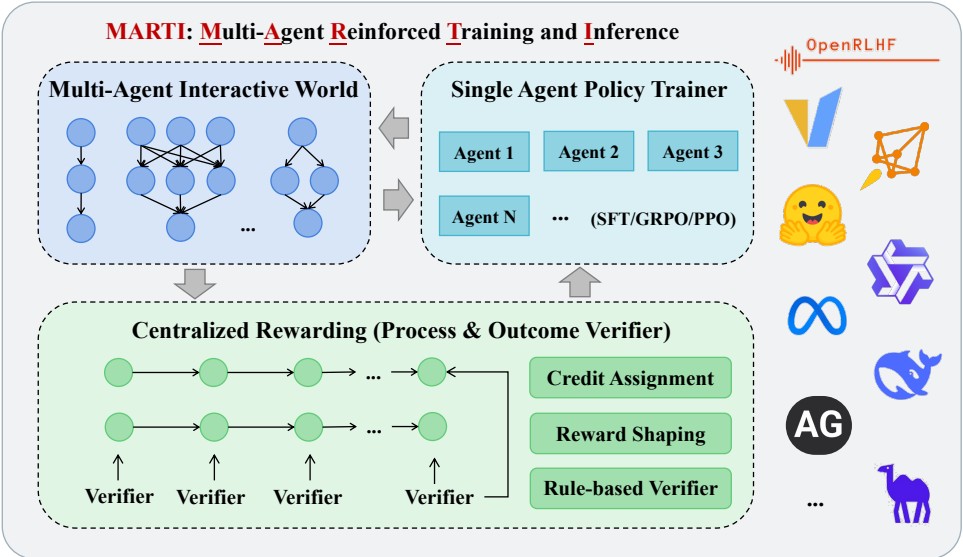

Figure 1: Overview and motivation behind of MARTI.

policy training is distributed across individual agents. As illustrated in Figure 1, MARTI consists of three core modules for rollout generation and policy training: Multi-Agent World, Centralized Reward Models, and Agent Policy Trainer. The relationships and detailed descriptions of each module are provided in the following section.

**Multi-Agent World.** This module serves as the environment for all multi-agent interactions and experience generation. Its core functions are to execute prompt-driven rollouts according to specified interaction workflows, manage the credit assignment mechanism for resulting trajectories, and perform the necessary data format conversion for downstream RL training of individual agents. To ensure maximal experimental flexibility, the system supports the dynamic injection of custom workflows, which define the multi-agent interaction logic and provide access to vLLM engines [1]. Trajectories collected in this environment are subsequently processed using Centralized Reward Models. A key architectural feature is the workflow's support for asynchronous generation, which significantly enhances data throughput. The abstract workflow interface is defined in Code 1, and a complete code example is provided in Appendix B.3 for reference.

**Centralized Reward Models.** Following world interactions, this module collects trajectories and performs credit assignment and reward shaping. Initial global rewards are computed using either rule-based strategies or generative reward models, which are then decomposed into agent-level rewards for subsequent agent training. Section 2.2.1 introduces rule-based rewards (e.g., DeepSeek-R1) and influence-aware reward shaping for MAS. For open-domain applications, Section 2.2.2 presents generative reward models that extend to LLM-as-judge approaches (Zheng et al., 2023; Gu et al., 2024) for multi-agent reward allocation across roles and collaborations.

**Agent Policy Trainer.** After trajectory collection and reward allocation, MARTI distributes agent-specific trajectories and rewards to individual policy trainers. Here, backbone LLMs undergo supervised fine-tuning or reinforcement learning. Section 2.2.3 discusses policy training strategies. Leveraging distributed training capabilities, we implement various RL algorithms from OpenRLHF, including REINFORCE++ (Hu, 2025), GRPO (Shao et al., 2024), and PPO (Schulman et al., 2017), while maintaining extensibility for novel algorithms such as PRIME (Cui et al., 2025). We additionally integrate supervised fine-tuning during on-policy rollout training to enhance stability and accelerate convergence. These dynamic training strategies warrant further investigation regarding on-policy and off-policy combinations (Yan et al., 2025; Tang et al., 2025).

---

[1] https://github.com/vllm-project/vllm

## 2.2 ALGORITHMS IMPLEMENTATION

In this section, we present the implementation details of reward allocation and policy training for multi-agent training in MARTI. For reward allocation, we first discuss rule-based reward shaping (Section 2.2.1), followed by generative reward models for open-domain applications (Section 2.2.2), and finally policy training strategies (Section 2.2.3).

### 2.2.1 RULE-BASED REWARD SHAPING

For mathematical problems with verifiable solutions, we employ rule-based reward models such as DeepSeek-R1 (Guo et al., 2025). This approach is particularly effective for mixture-of-agents (Wang et al., 2025a) and multi-agent debate (Du et al., 2024) scenarios, where each agent's output can be directly evaluated against the ground truth solution, enabling precise reward assignment based on predefined scoring rules. To improve temporal consistency and leverage historical information in multi-turn interactions, we introduce an inference-aware reward shaping strategy from MAPoRL (Park et al., 2025). This method integrates past performance estimates with current rewards. Specifically, the approach combines an immediate correctness reward from a task verifier with a dynamic adjustment derived from the agent's historical performance. This historical performance is calculated as the average reward across previous interactions.

We implement two variants: (1) a Quality Mode, which encourages consistency by aligning current performance with historical correctness, and (2) a Margin Mode, which directly rewards agents for surpassing their historical average performance. Additionally, two historical evaluation scopes are provided: one considers only the most recent interaction, offering immediate but potentially variable feedback, while the other averages across all past interactions for more stable and reliable estimates. These modular and flexible strategies effectively reduce overfitting to single-turn outcomes, enhancing long-term collaboration effectiveness in multi-turn scenarios.

Let $R_t^i \in [0, 1]$ denote the immediate correctness reward assigned by a task verifier for agent $i$ at turn $t$, and let $Q_t^i \in [0, 1]$ represent the historical performance estimate of the agent, computed over a set of previous interactions:

$$Q_t^i = \frac{1}{|\mathcal{H}_t^i|} \sum_{k \in \mathcal{H}_t^i} R_k^i, \tag{1}$$

where $\mathcal{H}_t^i \subset \{1, \ldots, t-1\}$ denotes the historical evaluation scope (e.g., most recent round or all previous rounds). We define the dynamic shaping term $\Delta_t^i$ under two modes:

$$\text{Margin Mode:} \quad \Delta_t^i = R_t^i - Q_t^i, \tag{2}$$

$$\text{Quality Mode:} \quad \Delta_t^i = Q_t^i \cdot R_t^i - (1 - Q_t^i)(1 - R_t^i). \tag{3}$$

The final shaped reward $\tilde{R}_t^i$ is then given by: $\tilde{R}_t^i = R_t^i + \alpha \cdot \Delta_t^i$ where $\alpha \in \mathbb{R}_{\geq 0}$ is a tunable hyperparameter controlling the influence of historical consistency.

### 2.2.2 GENERATIVE REWARD MODELS

Recent advances have demonstrated that LLMs can effectively evaluate response quality, enabling their use as generative reward models (GenRMs) to enhance policy model reasoning capabilities (Zhang et al., 2025e; Mahan et al., 2024; Zhao et al., 2025). Building on these developments, we implement GenRMs in MARTI for both verifiable and open-domain problems. Our framework supports GenRMs through either local vLLM engines or OpenAI-compatible APIs, with a defined GenRM that assigns scalar rewards to given problem-trajectory pairs.

Furthermore, we investigate specialized GenRMs for multi-agent systems (MAS) that explicitly address common failure modes identified in prior work (Cemri et al., 2025; Zhang et al., 2025f). These models show particular promise for improving collaborative behaviors in MAS. We continue to optimize this functionality, with further discussion reserved for future work.

### 2.2.3 POLICY MODEL TRAINING

Upon obtaining rollout experiences comprising individual trajectories and corresponding rewards for each agent, we initiate distributed training of agent policy models. The training leverages adapted

implementations from OpenRLHF, supporting various reinforcement learning algorithms including REINFORCE++ (Hu, 2025), GRPO (Shao et al., 2024), and PPO (Schulman et al., 2017). Notably, all agent policies are trained using identical RL algorithms to maintain consistency.

Furthermore, we augment the training process by incorporating additional imitation learning strategies during on-policy rollouts. These include supervised fine-tuning (SFT) and direct preference optimization (DPO) (Rafailov et al., 2023), extending beyond OpenRLHF's native capabilities. This integration enables dynamic selection of training strategies tailored to specific application requirements, such as stable training and faster convergence.

## 3 EXPERIMENTS

### 3.1 EXPERIMENTAL SETUP

**Datasets.** We utilize competition-level mathematical datasets for our experiments, including AIME24 (AI-MO, 2024a), AMC (AI-MO, 2024b), and MATH-500 (Lightman et al., 2024). All datasets are adapted from the publicly released DeepScaleR project materials [2].

**Models.** For non-reasoning models, we use `Qwen2.5-3B` and `Qwen2.5-3B-Instruct` (Yang et al., 2024). For reasoning models, we utilize `Qwen3-1.7B` (Team, 2025) and `DeepScaleR-1.5B-Preview` (Luo et al., 2025) for experiments. We also incorporate results for `Qwen2.5-7B/14B-Instruct`, `DeepSeek-R1-Qwen-7/14B`, and `OpenAI-o1-mini`.

**Inference Details.** For multi-agent workflow inference, we use a temperature of 0.6 and top_p of 0.95 for all models. The max generation token is set to 8192 for non-reasoning models and 16384 for reasoning models. For reasoning models with outputs like "`<think> reasoning </think><answer>final answer</answer>`", agents exclusively exchange final answers without their intermediate thinking processes with each other.

**Training Details.** We employ the MARTI framework to train both base and reasoning models, specifically `Qwen2.5-3B` and `DeepScaleR-1.5B-Preview`. For `Qwen2.5-3B`, we implement DeepSeek-R1 zero-like reinforcement learning training using Level 3-5 samples from the MATH dataset (Hendrycks et al., 2021) like previous works (Zeng et al., 2025; Liu et al., 2025b). The `DeepScaleR-1.5B-Preview` model, which exhibits strong inherent reasoning capabilities but presents training challenges, undergoes test-time reinforcement learning (TTRL) (Zuo et al., 2025) adaptation on AIME benchmark data. We maintain the same maximum generation tokens and temperature settings as used during inference, while extending the maximum prompt token length to 8192. For multi-agent reinforcement learning, we employ a cluster configuration consisting of 3 nodes, each equipped with 8 A800 80GB GPUs, allocating one full node per agent.

**Evaluation Metrics.** We evaluate model performance using accuracy scores computed for all datasets with open-source scripts from `Qwen2.5-Math`[3]. Additionally, we measure `Pass@1` by averaging scores across multiple responses and compute `Maj@N` (where $N = 4$ or 6) under the same inference budget using multi-agent reinforcement learning (RL). For conciseness, we abbreviate multi-agents debate, mixture-of-agents, and chain-of-agents as `MAD`, `MoA`, and `CoA`, respectively.

### 3.2 MAIN RESULTS

We present comparative results for both non-reasoning and reasoning models across different training and inference configurations in Figure 2 (instruction models) and Figure 3 (reasoning models), followed by a multi-perspective analysis:

**Failures of Multi-agent Workflows.** Our experimental results demonstrate that both non-reasoning and reasoning models achieve superior performance through majority voting compared to multi-agent workflows under equivalent computational budgets. This observation aligns with existing literature documenting failures in LLM-based multi-agent systems (Cemri et al., 2025; Zhang et al., 2025f;c), which identifies two key limitations: (1) inability to adhere to role specifications, and (2) failure to effectively utilize inter-agent interaction information. We attribute these shortcomings to the

---

[2]https://github.com/agentica-project/deepscaler
[3]https://github.com/QwenLM/Qwen2.5-Math

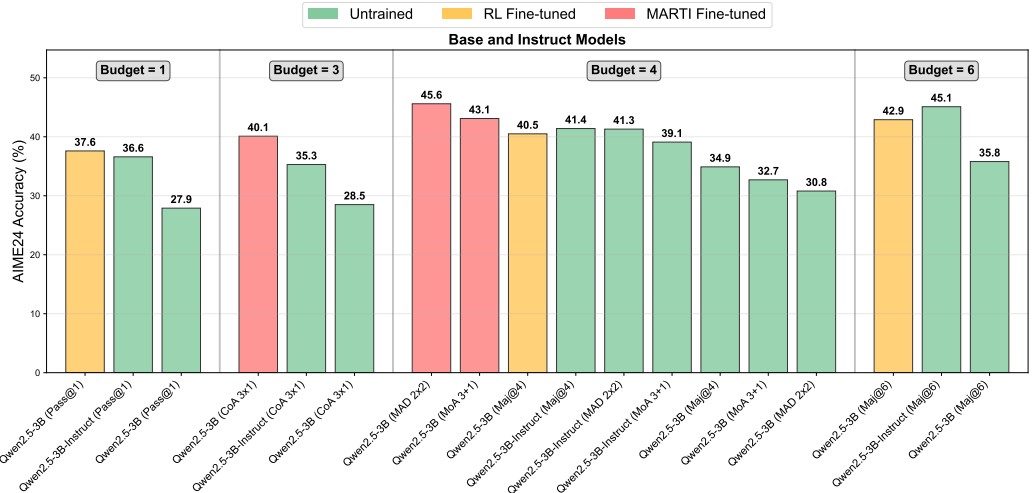

Figure 2: Average scores of Qwen2.5-3B base and instruct models under different budget and settings.

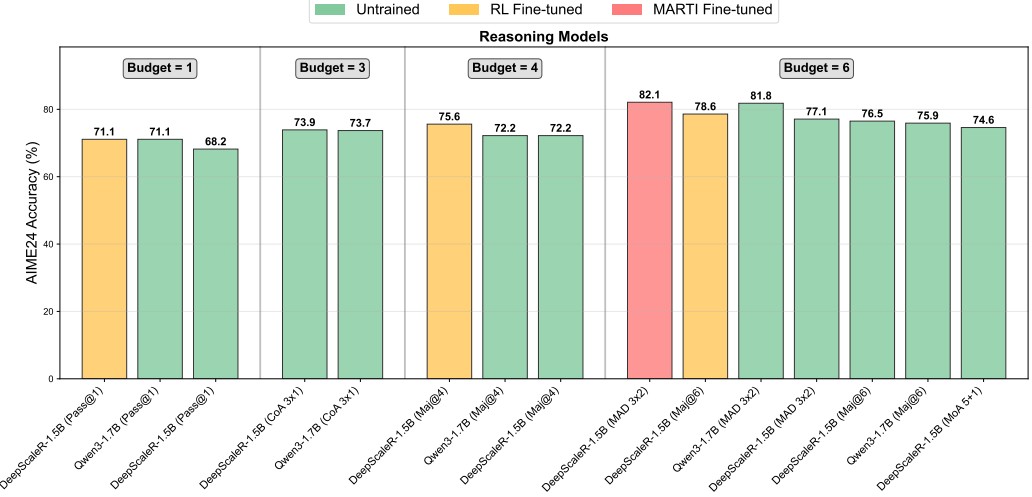

Figure 3: Average scores of reasoning models under different budget and settings.

predominant single-agent training paradigm of current LLMs, which inherently lacks exposure to multi-agent dynamics. These findings motivate our proposed MARTI framework, which implements MARL to develop advanced reasoning capabilities through structured agent interactions.

**MARTI Enhances Base Models Using Zero-like RL.** We further investigate training base models with zero-like RL using MARTI. For the `Qwen2.5-3B` model, we compare standard RL training with MARTI-based training and observe that multi-agent systems trained with MARTI achieve a higher performance upper bound than single-agent systems. Notably, multi-agent debate yields the best results under the same computational budget. Consistent with prior research, our results demonstrate that base models enhanced with reinforcement learning achieve comparable performance to instructed models. This finding further supports the established conclusion that RL primarily enhances a model's intrinsic pre-trained capabilities rather than imparting new knowledge. These findings suggest the need to explore novel reinforcement learning approaches that enhance individual model capabilities, such as through multi-agent interaction paradigms.

**MARTI Enhances Large Reasoning Models.** To explore the upper bound of multi-agent training, we apply test-time reinforcement learning (TTRL) to large reasoning models (`DeepScaleR-1.5B-Preview`). Our results demonstrate TTRL's effectiveness for LRMs, par-

ticularly on complex tasks. Notably, Multi-Agent Debates (MAD) achieve a score of 66.7 on AIME, significantly outperforming other same-cost configurations, including OpenAI-o1-mini.

**Multi-Agent RL Achieves a Higher Performance Upper Bound Than Single-Agent Systems.** After analyzing the performance of both base models and large reasoning models trained with MARTI, we find that multi-agent RL consistently achieves a higher performance upper bound than single-agent systems. This demonstrates that, under the same inference budget, reinforced multi-agent models attain superior benchmark scores compared to their single-agent counterparts. Furthermore, these results suggest that reinforced multi-agent training can enhance advanced reasoning capabilities, presenting a promising direction for future research in reasoning optimization.

Table 2: Results for Llama-3.2-3B-Instruct across various workflows and training configurations. Under an equivalent inference budget, MARTI consistently outperforms both single-agent reinforcement learning and majority-vote baselines.

| Llama-3.2-3B-Instruct | AIME | AMC | MATH500 | Avg |
|---|---|---|---|---|
| Single Agent (Pass@1) | 3.3 | 12.4 | 32.2 | 16.0 |
| + RL | 11.7 | 25.6 | 48.9 | 28.7 |
| Single Agent (Maj@4) | 6.6 | 18.1 | 36.6 | 20.4 |
| + RL | 11.7 | 27.7 | 50.6 | 30.0 |
| MAD 2×2 | 3.3 | 16.9 | 38.4 | 19.5 |
| + RL (MARTI) | 13.3 | 29.5 | 53.6 | 32.1 |
| MoA 3×1 | 6.6 | 16.9 | 37.2 | 20.2 |
| + RL (MARTI) | 11.7 | 28.7 | 52.6 | 31.0 |

Table 3: Comparison of REINFORCE++ (RF++) and GRPO on Qwen2.5-3B. Both algorithms produce strong performance gains; GRPO achieves marginally better results on most evaluated metrics.

| Qwen2.5-3B | AIME | AMC | MATH500 | Avg |
|---|---|---|---|---|
| Single-Agent + RF++ | 10.0 | 36.1 | 66.7 | 37.6 |
| Single-Agent + GRPO | 13.3 | 34.6 | 66.0 | 37.9 |
| MAD 2×2 + RF++ | 16.7 | 49.4 | 70.8 | 45.6 |
| MAD 2×2 + GRPO | 16.7 | 50.0 | 71.2 | 46.0 |

Table 4: Ablation study on reward shaping for Qwen2.5-3B. Removing reward shaping results in substantial performance degradation for both MAD and MoA architectures.

| Qwen2.5-3B | AIME | AMC | MATH500 | Avg |
|---|---|---|---|---|
| MAD 2×2 w/ reward shaping | 16.7 | 49.4 | 70.8 | 45.6 |
| MAD 2×2 w/o reward shaping | 6.6 | 36.6 | 66.7 | 36.6 |
| MoA 3×1 w/ reward shaping | 13.3 | 47.0 | 69.0 | 43.1 |
| MoA 3×1 w/o reward shaping | 10.0 | 38.9 | 65.4 | 38.1 |

## 3.3 ABLATION STUDIES

**Different Model Families** To examine whether MARTI generalizes beyond Qwen-based models, we apply both single-agent and multi-agent RL to the Llama-3.2-3B-Instruct backbone. Table 2 summarizes the results. Single-agent RL already brings a large improvement over the supervised Pass@1 baseline (from 16.0 to 28.7 on average). On top of this, applying MARTI to multi-agent workflows yields further gains: for example, MAD 2×2 with RL reaches an average score of 32.1, outperforming both single-agent RL (28.7) and majority-vote RL (30.0). MoA 3×1 with RL achieves a similar average score of 31.0. These trends are consistent across all three benchmarks, indicating that the benefits of MARTI are not tied to a specific model architecture.

**Different Algorithms** We further compare two policy-gradient estimators, REINFORCE++ (RF++) and GRPO, on Qwen2.5-3B. As shown in Table 3, both algorithms improve substantially over the supervised single-agent baseline, and GRPO provides a slight but consistent edge. For instance, MAD

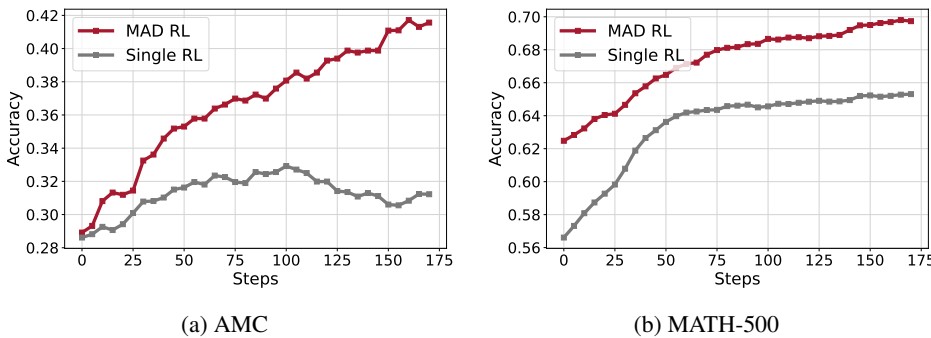

Figure 4: Accuracy of MAD (Qwen2.5-3B) on AMC and MATH-500

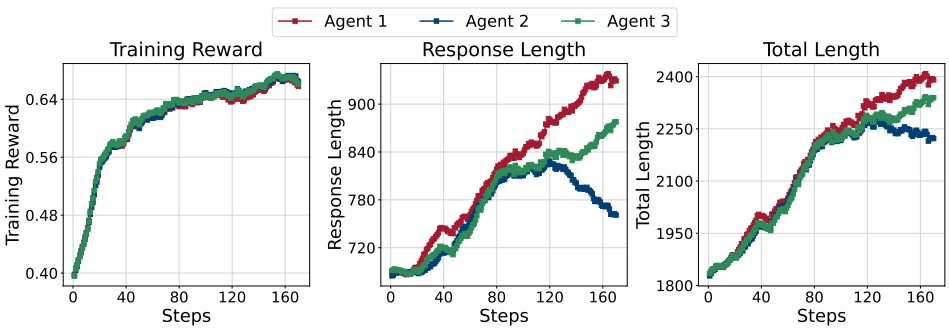

Figure 5: Training Dynamics of MAD (Qwen2.5-3B) with RL on MATH.

2×2 with GRPO achieves an average score of 46.0, compared to 45.6 with RF++. These results indicate that MARTI is robust to the specific choice of policy-gradient algorithm, and that the main gains arise from the multi-agent RL formulation itself.

**Reward Shaping.** MARTI employs a delta-style reward shaping mechanism that compares the current turn of an agent with its own historical performance, thereby rewarding relative improvements instead of only absolute correctness. Table 4 reports ablations on Qwen2.5-3B for MAD 2×2 and MoA 3×1. Removing reward shaping causes a clear drop in average performance: from 45.6 to 36.6 for MAD and from 43.1 to 38.1 for MoA. This shows that reward shaping is essential for stabilizing multi-agent RL; purely outcome-based rewards are more prone to instability and reward hacking in multi-turn interaction, whereas our shaping provides smoother optimization signals that better align with collaborative reasoning quality.

## 4 DISCUSSIONS

### 4.1 CASE STUDY: MULTI-AGENTS DEBATE

**Experimental Setup.** We conduct multi-agent debate training using two model architectures: `Qwen2.5-3B` and `DeepScaleR-1.5B-Preview`. The `Qwen2.5-3B` model is trained using REINFORCE++ on Level 3 to 5 samples from the MATH-500 dataset, while `DeepScaleR-1.5B-Preview` employs TTRL on the AIME benchmark.

**Training Dynamics.** Model accuracy results are presented for both AMC and MATH-500 benchmarks in Figures 4a and 4b, respectively. Additionally, we analyze the training dynamics of RL in Figure 5 and TTRL optimization in Figure 6.

We present additional case studies analyzing training dynamics across various multi-agent architectures in Appendix C, including Mixture-of-Agents (MoA) (Appendix C.1), Chain-of-Agents (CoA) (Appendix C.2), and Judge-based Debate (Appendix C.3).

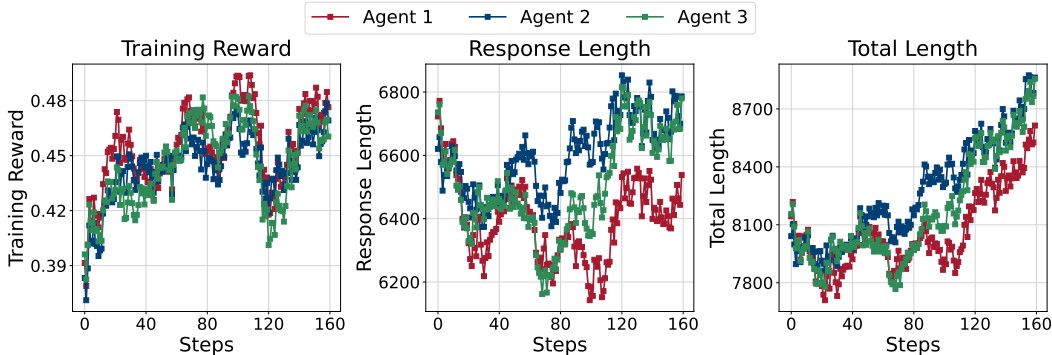

Figure 6: Training Dynamics of MAD (DeepScaleR-1.5B-Preview) with TTRL on AIME.

Table 5: Statistics of asynchronous rollouts in MARTI. (a) End-to-end rollout time vs. concurrency for Chain-of-Agents and MAD. (b) Total rollout time vs. number of interaction rounds for different concurrency settings.

**(a) Time vs. concurrency (seconds)**

| Workflow | Sync | Async×32 | Async×64 | Async×128 | Async×256 | Async×512 |
|----------|------|----------|----------|-----------|-----------|-----------|
| Chain | 612.6 | 615.5 | 553.5 | 508.5 | 515.4 | 498.4 |
| MAD | 593.5 | 732.9 | 616.8 | 592.4 | 569.4 | 561.2 |

**(b) Time vs. interaction rounds (seconds)**

| round | Sync | Async×32 | Async×64 |
|-------|------|----------|----------|
| 2 | 916.6 | 1074.1 | 887.5 |
| 5 | 2194.0 | 2186.1 | 2022.8 |
| 8 | 3308.0 | 3004.4 | 2928.0 |

## 4.2 STATISTICS OF ASYNCHRONOUS GENERATIONS

Previous rollouts in RL frameworks are typically performed in batches for batch generation. However, this approach proves inefficient for multi-turn interactions, such as multi-turn tool calls and multi-agent interactions, due to significant discrepancies in time costs during generation. As a result, asynchronous generation has become a core feature in mainstream RL frameworks, particularly in agentic RL systems such as OpenRLHF (Hu et al., 2024a), veRL (Sheng et al., 2024), and AReaL (Fu et al., 2025). To the best of our knowledge, MARTI is the first framework to support asynchronous generation for multi-turn, multi-agent scenarios.

We further analyze the effect of asynchronous rollouts in MARTI. Table 5 summarizes the time costs for synchronous and asynchronous execution in both Chain-of-Agents and Multi-Agent Debate workflows under different levels of concurrency and interaction depth. With moderate concurrency, asynchronous rollouts consistently reduce end-to-end inference time, but very large numbers of parallel workers become compute bound and yield diminishing returns. When the interaction depth is small (e.g., 2 rounds), trajectories are short and synchronization overhead is negligible, so Async×64 provides only a modest ∼ 3% speed-up over the synchronous baseline. As the number of rounds increases, rollout latency grows and the throughput advantage of asynchrony becomes more pronounced. Overall, asynchronous generation is most beneficial for deep, interaction heavy workflows, while shallow workflows are primarily limited by raw compute rather than synchronization.

## 5 CONCLUSION

We present MARTI, a unified framework integrating multi-agent reinforcement learning (RL) with inference for LLM-based systems. By combining scalable RL training (via OpenRLHF) with adaptive multi-agent workflows, MARTI outperforms single-agent TTRL in reasoning tasks, achieving

advanced performance on AIME. Challenges like reward modeling and real-world deployment persist, but MARTI advances MAS capabilities through built-in credit assignment and support for diverse reward models. Our work demonstrates that multi-agent RL elevates performance ceilings beyond single-agent approaches, offering a pathway to enhance reasoning in practical applications. Future work will focus on optimizing MAS training for broader adoption.

## ETHICS STATEMENT

This work presents MARTI, a framework for LLM-based multi-agent reinforcement learning and inference. We use established public benchmarks to ensure transparent and unbiased evaluation, while minimizing computational waste through efficient configurations.

## REPRODUCIBILITY STATEMENT

We provide comprehensive details to ensure reproducibility, including implementation specifics in Section 3.1 (models, inference details, training procedures, and evaluation metrics). Additionally, the anonymous MARTI codebase is provided in `https://github.com/TsinghuaC3I/MARTI`.

## ACKNOWLEDGMENTS

This work is supported by the National Science and Technology Major Project (2023ZD0121403), Young Elite Scientists Sponsorship Program by CAST (2023QNRC001), and National Natural Science Foundation of China (No. 62406165). We thank anonymous reviewers for their insightful comments and suggestions.

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

# A RELATED WORKS

## A.1 MULTI-AGENT LLM SYSTEMS

Previous researches explore multi-agent LLM workflows, focusing on multi-agent debate (Du et al., 2024; Liang et al., 2024; Xu et al., 2023; Yin et al., 2023; Wang et al., 2024; Chen et al., 2024b; Zhang et al., 2025c), communication topology (Chan et al., 2024; Chen et al., 2024c; Zhuge et al., 2024; Qian et al., 2025; Li et al., 2024; Yue et al., 2025b; Zhang et al., 2025b;a), and test-time scaling (junyou li et al., 2024; Wang et al., 2025a; Antoniades et al., 2024; Ye et al., 2024; Yang et al., 2025). These works demonstrate the potential of multi-agent systems to enhance collaborative problem-solving and scalability, effectively managing complex interactions across diverse tasks.

Multi-agent frameworks improve collaborative task-solving effectively (Li et al., 2023; Chen et al., 2024a; Liang et al., 2025). CAMEL (Li et al., 2023) uses a role-playing framework where a user agent decomposes tasks and an assistant agent executes them, guided by an inception prompt. MetaGPT (Hong et al., 2024) simulates the collaboration of a software company by assigning distinct roles for handling complex tasks. However, fixed agent roles restricts the adaptability of multi-agent frameworks. To address this, AutoAgents (Chen et al., 2024a) dynamically generates specialized agents and coordinates them through a central planning module for complex tasks. AutoGen (Wu et al., 2023a) focuses on developing LLM applications through layered and extensible multi-agent design. Similarly, OpenManus (Liang et al., 2025) provides a modular framework with agents, flows, prompts, and tools, adopting a tool-centric ReAct (Yao et al., 2023) paradigm to support plan-then-act decision-making, effectively handling tasks requiring extended reasoning.

Despite the wide variety of existing multi-agent LLM systems, significant performance gains at the emergent level remain elusive (Liu et al., 2019; Chen et al., 2024c). In some tasks, multi-agent frameworks exhibit only marginal gains over single-agent approaches (Pan et al., 2025). These limitations often stem from the inherent constraints of single LLM agents. When handling context, a single agent may fail to follow task or role instructions (Wen et al., 2024; Wang et al., 2025b), or lose focus in long-context scenarios (Zhang et al., 2024b; An et al., 2024). Additionally, the model itself may generate outputs with factual hallucinations or misinterpret contextual cues (Zhang et al., 2023; Jiang et al., 2024). On the other hand, the design of the workflow and inter-agent coordination mechanisms often plays a critical role in MAS failures. Common issues include overly complex system design (Kapoor et al., 2024), disorganized memory management (Han et al., 2024), and failure of verify-refine mechanisms (Huang et al., 2024). Our proposed MARTI framework provides a platform for testing, observing, and mitigating such failures through training.

## A.2 REINFORCEMENT LEARNING FOR LLMS

Test-time scaling (TTS) is designed to enhance the capabilities of LLMs in handling complex tasks by increasing computational resources at the time of testing. Prior research (Snell et al., 2024; Liu et al., 2025a) indicates that TTS is more efficient than scaling during pre-training (Kaplan et al., 2020); thus, reallocating the same computational resources from pre-training to test-time could yield greater improvements in model performance. Current studies on TTS fall into two categories (Welleck et al., 2024): parallel generation and sequential generation. Parallel generation entails LLMs producing multiple candidate responses (self-consistency (Wang et al., 2022; Chen et al., 2023), best-of-N (Stiennon et al., 2020; Nakano et al., 2021)), decision steps (Monte Carlo Tree Search (Zhou et al., 2023; Xie et al., 2024)), or tokens (Reward-guided Search (Deng & Raffel, 2023; Khanov et al., 2024)) during inference. Subsequently, an aggregation strategy is applied to integrate these candidates, commonly utilizing process reward models (Lightman et al., 2024; Wang et al., 2023a; Zhang et al., 2025d). Concurrently, sequential generation focuses on extending the LLMs' output to include longer responses with reflective and chain-of-thought processes (Wei et al., 2022; Madaan et al., 2023). Although prompting techniques are widely adopted, they are often constrained by the capabilities of the underlying models. Notably, DeepSeek-R1 (Guo et al., 2025) represents a significant advancement in this area, achieving extended reasoning capabilities in pre-trained language models through outcome-based RL, like group relative policy optimization (Shao et al., 2024). Compared to the first approach, which requires intensive process-level supervision (Yuan et al., 2024), the second approach is more scalable due to its reliance on rule-based rewards.

### A.3 MULTI-AGENT REINFORCEMENT LEARNING

Multi-agent reinforcement learning has emerged as a powerful framework for modeling strategic interactions, guided by game-theoretic principles that shape both learning dynamics and reasoning processes (Yang, 2021; Huh & Mohapatra, 2023). Recent research has focused on addressing its unique challenges such as non-stationarity, credit assignment, and scalability. Wang et al. (2023b) introduce a sequential agent-wise update scheme with off-policy correction, ensuring monotonic improvement and enhancing performance in cooperative tasks. Slumbers et al. (2023) leverage shared policies, centralized training, and natural language communication to enhance performance in text-based environments. Zhang et al. (2024a) shows that LLM-driven agents with Theory of Mind improve perceived coordination in human-AI teams, though bidirectional communication can hinder performance. Wan et al. (2025) separates meta-thinking and reasoning into distinct agents, achieving improved generalization and performance on complex reasoning tasks. Park et al. (2025) jointly trains multiple LLMs via inference-aware rewards to foster effective, transferable collaboration in multi-turn tasks. Lu et al. (2025) proposes a preference-guided multi-agent federated framework that integrates rule-based models and human preference signals in urban autonomous driving scenarios. Thind et al. (2025) translates natural language optimization problems into executable solvers through role-specialized agents.

## B WORKFLOWS

### B.1 WORKFLOW CODE

Listing 1: The pseudo-code of the Abstract Workflow.

```python
async def workflow(
    prompt: str,
    label: str,
    agents: List[Dict[str, Any]],
    tool_manager,
    task: str,
    metadata: Optional[Dict] = None,
    **kwargs
) -> Dict[str, Any]:
    # Customized Interactions
    trajectory = [
        {
            "turn_id": 0,
            "agent_index": 0,
            "agent_name":  "agent0",
            "agent_role": "generator",
            "agent_input": "input_example",
            "agent_output": "output_example",
            "metadata": {}
        },
        # Add more turns
    ]
    rewards = [0]
    # Add reward for each turn if exist

    return {
        "prompt": prompt,
        "label": label,
        "trajectory": trajectory,
        "final_reward": rewards[-1]
    }
```

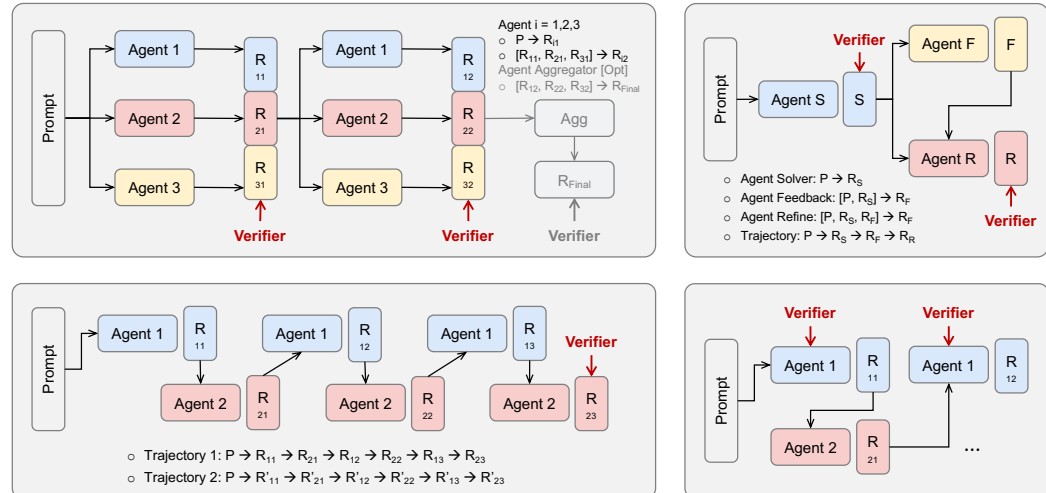

Figure 7: MAS Examples for Typical Multi-Agent Workflows.

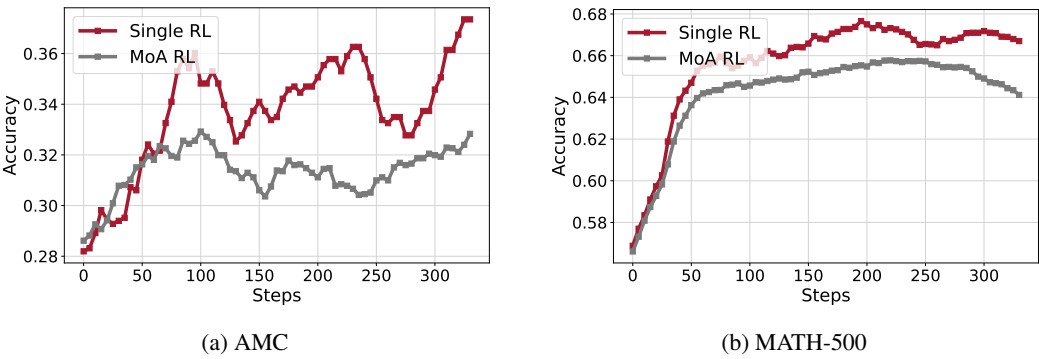

(a) AMC                                        (b) MATH-500

Figure 8: Accuracy of MoA (Qwen2.5-3B) on AMC and MATH-500

## B.2 WORKFLOW EXAMPLE

We introduce and compare several workflows in Figure 7, including mixture-of-agents and chain-of-agents. For the chain-of-agents workflow, two typical credit assignment strategies are considered: (1) assigning verifiable rewards at each turn or (2) assigning the final reward at the end, with the final reward distributed across the intermediate turns. These workflows are fully supported in the MARTI framework for further experimentation.

## B.3 CODE EXAMPLE

A full example for MathChat with three agents is provided in Figure 2.

## C CASE STUDY

### C.1 CASE 1: MIXTURE-OF-AGENTS

**Experimental Setup.** We evaluate a mixture-of-agents approach using the `Qwen2.5-3B` model, trained on Levels 3 through 5 of the MATH-500 training dataset.

**Training Dynamics.** The model's accuracy results are presented for both AMC and MATH-500 benchmarks in Figures 8a and 8b, respectively. Furthermore, we analyze the complete training dynamics in Figure 9, including training rewards, response length, and total length.

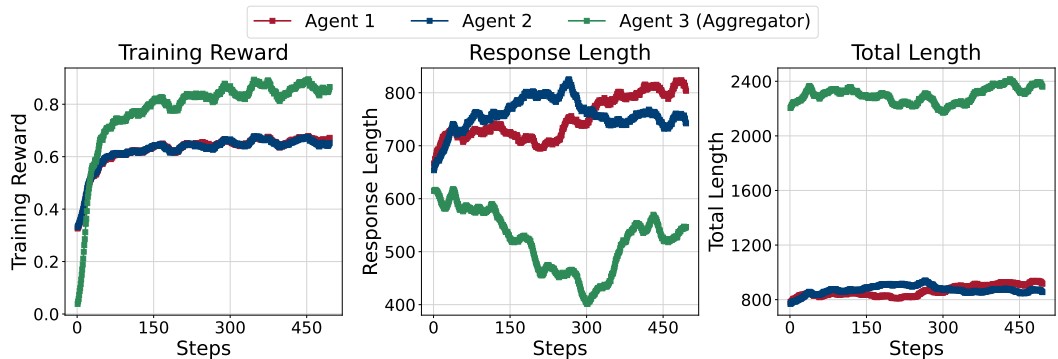

Figure 9: Training Dynamics of MoA (Qwen2.5-3B) with RL on MATH.

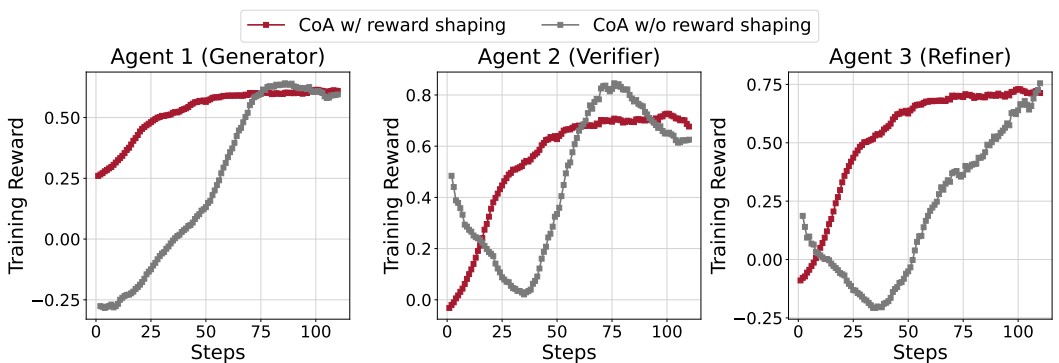

Figure 10: Training Rewards of CoA (Qwen2.5-3B) with RL on MATH.

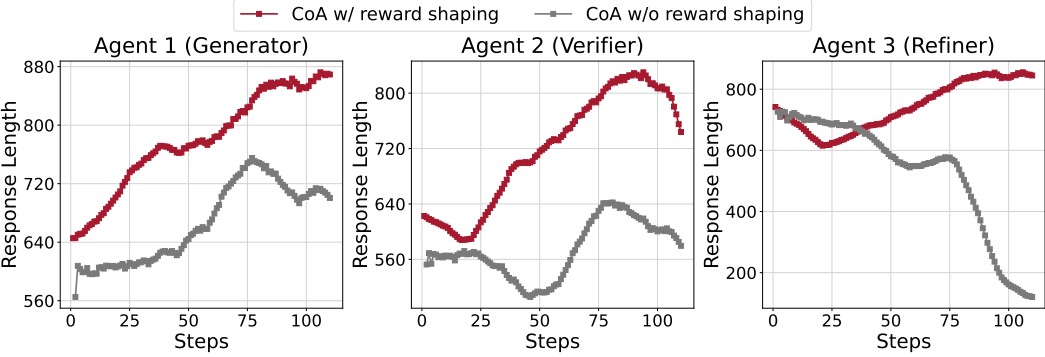

Figure 11: Training Response Length of CoA (Qwen2.5-3B) with RL on MATH.

## C.2   CASE 2: CHAIN-OF-AGENTS

**Experimental Setup.** We investigate chain-of-agents reinforcement learning (RL) using Levels 3–5 of the MATH-500 training set. Our evaluation compares standard RL training with a quality-aware reward shaping variant to assess performance improvements.

**Training Dynamics.** The training process is characterized by three key metrics:

- Training reward curve in Figure 10.

- Response length dynamics in Figure 11.

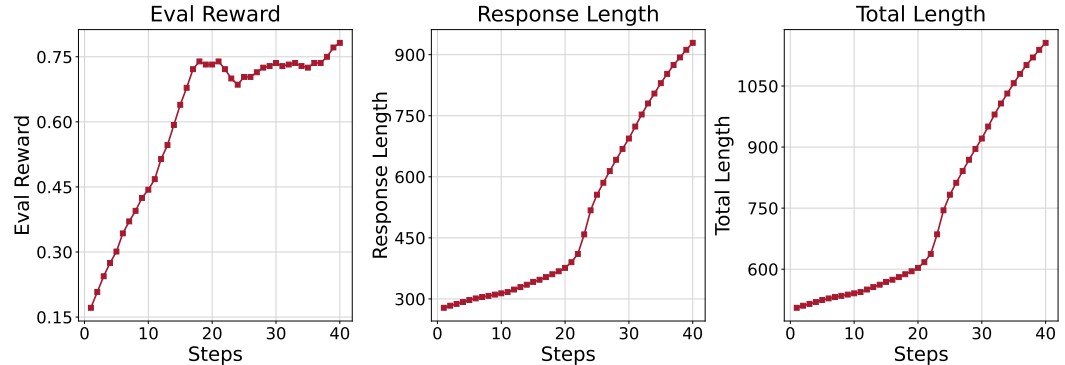

Figure 12: Training Dynamics of Judge-based Two-player Debate.

Table 6: Average number of output tokens per instance on Qwen2.5-3B across tasks and workflows. Multi-agent workflows operate under a comparable token budget to Majority@4, and can even be more efficient.

| Output Tokens | AIME | AMC | MATH500 | Avg |
|---|---|---|---|---|
| Single-Agent (Avg@4) | 4532 | 3706 | 2322 | 3520 |
| MAD 2×2 | 2698 | 4268 | 2698 | 3221 |
| MoA 3×1 | 3083 | 2146 | 1518 | 2249 |

## C.3 CASE 3: JUDGE-BASED DEBATE

**Experimental Setup.** We conduct LLM training using GenRM-generated feedback under a debate setting [4]. Specifically, we use the round-robin tournament data from and train a Llama-3.2-1B-Instruct model to debate against a Llama-3.1-8B-Instruct model. The reward is the win rate in the round-robin tournament judged by a Qwen2.5-14B-Instruct model. **Training Dynamics.** We present the evaluation rewards, response length, and total trajectory length during training in Figure 12.

## D THE USE OF LARGE LANGUAGE MODELS

All core research ideas, theoretical derivations, experimental designs, and algorithmic innovations were developed by the authors without LLM assistance. Additionally, all paragraphs in the paper were originally written by humans. LLMs were used solely to fix bugs in the MARTI framework, under human review, and to polish sections of the paper.

## E COMPUTE ACCOUNTING AND INFERENCE BUDGET

**Unified Definition of Inference Budget.** To ensure fair comparisons between single- and multi-agent workflows, we standardize the *inference budget* in terms of the number of model rollouts under a fixed sampling configuration (temperature, top-$p$, maximum length, etc.). Concretely, a single-agent Majority@4 evaluation and a MAD 2×2 session both generate four trajectories and therefore consume an equivalent rollout budget. Similarly, MoA 3×1 produces three trajectories plus one final aggregation step, which in practice is comparable to Majority@4 in terms of compute.

We also report untrained multi-agent baselines (with the backbone kept frozen) to disentangle the benefit of test-time compute from that of learned collaboration. These baselines only change the interaction topology (e.g., voting, MAD, MoA) while keeping the rollout budget fixed, and thus highlight that the gains of MARTI mainly come from reinforcement learning on collaborative behaviours rather than simply sampling more trajectories.

---

[4] https://github.com/brendanhogan/DeepSeekRL-Extended

**Token-Level Statistics Across Workflows.** In addition to rollout counts, we measure the average number of output tokens per instance for different workflows on Qwen2.5-3B. Table 6 summarizes the results. MAD $2\times2$ consumes a similar number of tokens to the single-agent Majority@4 baseline (within roughly $10\%$ on average), while MoA $3\times1$ is even more token-efficient. This confirms that the accuracy gains reported in the main paper do not arise from substantially increased generation cost.

Listing 2: The pseudo-code of MathChat workflow.

```python
from typing import Dict, List, Any

async def workflow(
    prompt: str,
    label: str,
    agents: List[Dict[str, Any]],
    tool_manager: Any,
    task: str,
    **kwargs
) -> Dict[str, Any]:
    """
    Orchestrates an asynchronous multi-agent workflow and collects data
        for training.

    This example defines a three-step interaction:
    1. A 'generator' agent proposes a solution.
    2. A 'coder' agent implements the solution in code, which is then
        executed.
    3. A 'refiner' agent verifies all outputs to provide a final answer.

    The collected 'trajectory' retains all inputs, outputs, and rewards,
    forming a complete data sample for reinforcement learning.
    """
    # 1. Initialize workflow and identify agents by their predefined
        roles
    trajectory = []
    generator_agent, coder_agent, refiner_agent = agents[0], agents[1],
        agents[2]

    # --- Turn 1: Generator proposes a solution ---
    generator_input = f"Problem: {prompt}\nPlease reason step by step..."
    generator_response = await generator_agent["llm"].generate_async.
        remote(
        generator_input, generator_agent["sampling_params"]
    )
    generator_output = generator_response.outputs[0].text
    trajectory.append({
        "agent_role": "generator", "agent_input": generator_input, "
            agent_output": generator_output
    })

    # --- Turn 2: Coder writes and executes code based on the generator's
        solution ---
    coder_input = f"Problem: {prompt}\nSolver Output: {generator_output}\
        nWrite Python code..."
    coder_response = await coder_agent["llm"].generate_async.remote(
        coder_input, coder_agent["sampling_params"]
    )
    coder_output = coder_response.outputs[0].text

    # Use the tool manager to execute the generated code
    code_to_execute = extract_code(coder_output)
    execution_result, _ = await tool_manager.execute_tool(
        "code_interpreter", {"code": code_to_execute}
    )

```

```
49        trajectory.append({
50            "agent_role": "coder", "agent_input": coder_input, "agent_output"
                  : coder_output,
51            "metadata": {"tool_output": execution_result}
52        })
53
54        # --- Turn 3: Refiner verifies all outputs to produce a final answer
              ---
55        refiner_input = (f"Problem: {prompt}\nSolver Output: {
              generator_output}\n"
56                         f"Code Output: {execution_result}\nVerify and
                               provide the final answer...")
57        refiner_response = await refiner_agent["llm"].generate_async.remote(
58            refiner_input, refiner_agent["sampling_params"]
59        )
60        refiner_output = refiner_response.outputs[0].text
61        trajectory.append({
62            "agent_role": "refiner", "agent_input": refiner_input, "
                  agent_output": refiner_output
63        })
64
65        # 2. Evaluate the completed trajectory to assign rewards for RL
              training
66        all_outputs = [turn["agent_output"] for turn in trajectory]
67        all_rewards = auto_verify(task, all_outputs, [label] * len(
              all_outputs))
68
69        for turn, reward in zip(trajectory, all_rewards):
70            turn["agent_reward"] = reward
71
72        # 3. Return the structured data sample in the required format
73        return {
74            "prompt": prompt,
75            "label": label,
76            "trajectory": trajectory,
77            "final_reward": all_rewards[-1]
78        }
```

