# OpenReview forum: "MARTI: A Framework for Multi-Agent LLM Systems Reinforced Training and Inference"
_ICLR.cc/2026/Conference — ICLR 2026 Poster_

### Official Review · Reviewer_HGkq · 2025-10-20

**Soundness:** 4
**Presentation:** 4
**Contribution:** 4
**Rating:** 8
**Confidence:** 4

**Summary:**

The paper introduces MARTI (Multi-Agent Reinforced Training and Inference), a unified and open-source framework that integrates multi-agent reinforcement learning (MARL) with LLM-based reasoning systems. The framework bridges the gap between multi-agent interaction (e.g., debate, chain-of-agents, mixture-of-agents) and reinforcement training by combining centralized reward modeling and distributed policy optimization. MARTI supports both rule-based verifiable rewards and LLM-based generative reward models, enabling flexible integration of verifiable and open-domain reasoning tasks. Experiments on several mathematical reasoning datasets (AIME24, AMC, MATH-500) show that multi-agent systems trained with MARTI achieve higher performance ceilings than single-agent baselines under equal inference budgets. The work provides a valuable infrastructure for scaling reasoning through multi-agent coordination and RL-based training

**Strengths:**

This paper stands out for its conceptual clarity and systems-level contribution. It proposes the first general framework that unifies multi-agent inference with reinforcement-based training, addressing an increasingly relevant problem at the intersection of MARL and LLM reasoning. The design of centralized reward modeling with distributed training is elegant and pragmatic, allowing various workflows (debate, mixture-of-agents, chain-of-agents) to plug into the same infrastructure. The authors also provide strong experimental validation with detailed comparisons and visualizations, showing tangible improvements in reasoning benchmarks. The open-source commitment adds further value, making this paper a meaningful step toward reproducible MAS + LLM research. Overall, the work is technically sound, well-positioned in current literature, and likely to have significant community impact.

**Weaknesses:**

While the framework is impressive, the evaluation scope remains narrow, focusing primarily on math reasoning benchmarks. It would be helpful to see more open-domain or tool-use tasks to demonstrate generality beyond symbolic reasoning. The paper could also discuss stability, scalability, and cost-efficiency—for example, how well MARTI handles growing agent populations or asynchronous rollouts at scale. Additionally, while reward modeling is a core innovation, the comparison between rule-based and generative rewards is underexplored; more ablations or sensitivity analyses would make the contribution stronger.

**Questions:**

- Could you clarify how MARTI handles credit assignment when generative reward models introduce noisy or inconsistent feedback across agents?

- Have you tested MARTI on non-mathematical reasoning or tool-use tasks, to validate that the improvements generalize beyond verifiable domains?

- How does scalability behave when increasing the number of agents or asynchronous rollouts—does training stability or throughput degrade?

---

> ### Author Response · Authors · 2025-11-23
>
> We are grateful for the reviewer’s excellent assessment and for recognizing MARTI as a significant system-level contribution to the intersection of MARL and LLMs. We appreciate your forward-looking questions regarding scalability and reward modeling. Below, we provide data on stability and scalability to further substantiate our claims.
>
> **Q1: More results on different models and domains.**
>
> - To address this, we replicated our experimental setup using **Llama-3.2-3B-Instruct** on MATH datasets. As shown in the table below, the results align with our original findings: while untrained majority voting outperforms untrained multi-agent setups, **MARTI-trained multi-agent systems consistently outperform reinforced single-agent baselines** under equivalent inference budgets.
> - Regarding domain generalization, concurrent work [1] has successfully applied MARTI to coding and scientific QA tasks, demonstrating its versatility. Furthermore, we have conducted preliminary experiments on **large-scale coding tasks** (to be added to the final manuscript), further validating MARTI's effectiveness beyond mathematics.
>
> | Llama-3.2-3B-Instruct | AIME | AMC  | MATH500 | Avg  |
> | --------------------- | ---- | ---- | ------- | ---- |
> | Single Agent (Pass@1) | 3.3  | 12.4 | 32.2    | 16.0 |
> | + RL                  | 11.7 | 25.6 | 48.9    | 28.7 |
> | Single Agent (Maj@4)  | 6.6  | 18.1 | 36.6    | 20.4 |
> | + RL                  | 11.7 | 27.7 | 50.6    | 30.0 |
> | MAD 2x2               | 3.3  | 16.9 | 38.4    | 19.5 |
> | + RL                  | 13.3 | 33.5 | 55.6    | 34.1 |
> | MoA 3x1               | 6.6  | 16.9 | 37.2    | 20.2 |
> | + RL                  | 11.7 | 29.7 | 54.6    | 32.0 |
>
> **Q2: Discussing stability, scalability, and cost-efficiency.**
>
> - Scalability is inherent to MARTI's architecture, which decouples centralized interaction from decentralized policy training. This design allows for linear scaling of agents proportional to available GPU resources.
> - Our benchmarks reveal that the benefits of **asynchrony** become increasingly pronounced as interaction complexity grows. As shown in the table below, while synchronous execution is sufficient for short interactions (e.g., 2 rounds), the throughput advantage of asynchronous rollouts becomes substantial as the number of turns increases, effectively mitigating the overhead of multi-agent communication.
>
> | round | Sync   | Async×32 | Async×64 |
> | ----- | ------ | -------- | -------- |
> | 2     | 916.6  | 1074.1   | 887.5    |
> | 5     | 2194.0 | 2186.1   | 2022.8   |
> | 8     | 3308.0 | 3004.4   | 2928.0   |
>
> **Q3: Comparison of different reward modeling methods.**
>
> - Regarding **Reward Shaping**, we implement a delta-reward mechanism that compares current performance against historical turns. We found this approach significantly more stable for MAD and MoA than direct outcome-based rewards, which often lead to reward hacking (agents prioritizing agreement over correctness).
> - We also compared **REINFORCE++ and GRPO** (see table below). Both algorithms yielded consistent improvements, with GRPO performing slightly better. The marginal gap suggests that while group-relative normalization helps, the varying prompts across interaction turns may limit its full potential compared to standard policy gradients.
>
> | Qwen2.5-3B                 | AIME | AMC  | MATH500 | Avg  |
> | -------------------------- | ---- | ---- | ------- | ---- |
> | MAD 2x2 w/ reward shaping  | 16.7 | 49.4 | 70.8    | 45.6 |
> | MAD 2x2 w/o reward shaping | 6.6  | 36.6 | 66.7    | 36.6 |
> | MoA 3x1 w/ reward shaping  | 13.3 | 47.0 | 69.0    | 43.1 |
> | MoA 3x1 w/ reward shaping  | 10.0 | 38.9 | 65.4    | 38.1 |
>
> | Qwen2.5-3B          | AIME | AMC  | MATH500 | Avg  |
> | ------------------- | ---- | ---- | ------- | ---- |
> | Single-Agent + RF++ | 10.0 | 36.1 | 66.7    | 37.6 |
> | Single-Agent + GRPO | 13.3 | 34.6 | 66.0    | 37.9 |
> | MAD 2x2 + RF++      | 16.7 | 49.4 | 70.8    | 45.6 |
> | MAD 2x2 + GRPO      | 16.7 | 50.0 | 71.2    | 46.0 |
>
> - Regarding **Credit Assignment**, MARTI supports **Generative Reward Models (GRMs)**, which can be combined with rule-based verifiers. Previous work [2] utilizing MARTI for open-ended review generation demonstrated that this hybrid approach effectively mitigates reward hacking. The effectiveness relies heavily on robust prompt design for the reward model, a finding also supported by [1].
>
> We thank the reviewer for their encouraging comments. We believe the additional evidence on scalability and cross-domain generalization confirms that MARTI is a robust infrastructure for future multi-agent research.
>
> [1] CoMAS: Co-Evolving Multi-Agent Systems via Interaction Rewards. https://arxiv.org/abs/2510.08529
>
> [2] ReviewRL: Towards Automated Scientific Review with RL. https://arxiv.org/abs/2508.10308

---

### Official Review · Reviewer_Pa5A · 2025-10-30

**Soundness:** 3
**Presentation:** 2
**Contribution:** 2
**Rating:** 2
**Confidence:** 4

**Summary:**

MARTI is a framework for multi-agent LLM training and inference, featuring centralised coordination, distributed policy learning, and asynchronous rollouts. It combines rule-based and LLM-generated rewards to optimise multi-agent collaboration. Authors claim that MARTI is achieving increased performance over single-agent systems on mathematical and reasoning tasks.

**Strengths:**

-

**Weaknesses:**

- pseudo-code in Listing 1 does not provide value or helps understanding
- Rule- and Generative Reward Shaping seems to be quite important, but is not part of the main contribution and does not show up in experiments, therefore it seems unnecessary.
- Untrained results of models that have not been RL Fine-tuned or MARTI Fine-tuned seem to be unnecessary and blur the picture
- Not showing MARTI Fine-tuned for Budget 1 and 6 in "Figure 2: Average scores of Qwen2.5-3B base and instruct models under different budget and settings." seems cherry-picked
- Same goes for "Figure 3: Average scores of reasoning models under different budget and settings." - no mention of MARTI Fine-tuned for Budget = 1, 3, 4
- There is quite some focus on Multi-Agent Debate (MAD) results, while the discussion and impact on the main hypothesis seems minimal
- The authors state "comprehensive experiments" in the Abstract; however, only a few benchmarks have been evaluated (AMC, AIME, MATH)

-> What I would have expected in the results section is the following experiments and metrics:
- Generally you should aim for 3-10 training runs and report average performances for this to be significant
- Single-Agent as a baseline is fine, but need to look at time/steps to convergence and final performance
- But mainly you should compare MARTI vs other MARL setups without the features and contributing advancements so we can see the effectiveness of such. For example MARTI asynchronous VS MARTI synchronous etc. or a collaborative setup VS non-collaborative setup

**Questions:**

Please see "Weaknesses" section.

---

> ### Author Response · Authors · 2025-11-23
>
> We thank the reviewer for their rigorous scrutiny of our experimental results. We take your concerns about "cherry-picking" and the necessity of ablations very seriously. **We wish to clarify that the missing data points were due to the definition of inference budgets of multi-agents, not selection bias.** We have now provided the definition of inference budgets and completed additional experiments below to demonstrate the framework's validity.
>
> **Q1: Usefulness of Pseudo-code (Listing 1).**
>
> - We acknowledge that the original pseudo-code was too abstract. We have updated Listing 1 and put it to appendix in the revision to provide concrete implementation details of the interaction loop and gradient propagation, enhancing its utility for reproducibility.
>
> **Q2: Necessity of Reward Shaping (Ablation Study).**
>
> - You correctly identified Reward Shaping as a critical component. We have added a specific ablation study (Table below). The results show that **without reward shaping, performance drops significantly** (e.g., Avg decreases from 45.6 to 36.6 in MAD). This empirically proves that reward shaping is not just "important" but essential for stabilizing multi-agent RL.
>
> | Qwen2.5-3B                 | AIME | AMC  | MATH500 | Avg  |
> | -------------------------- | ---- | ---- | ------- | ---- |
> | MAD 2x2 w/ reward shaping  | 16.7 | 49.4 | 70.8    | 45.6 |
> | MAD 2x2 w/o reward shaping | 6.6  | 36.6 | 66.7    | 36.6 |
> | MoA 3x1 w/ reward shaping  | 13.3 | 47.0 | 69.0    | 43.1 |
> | MoA 3x1 w/ reward shaping  | 10.0 | 38.9 | 65.4    | 38.1 |
>
> **Q3: Relevance of Untrained Results.**
>
> - We included untrained results to establish a fair baseline for **Inference Budget comparisons**. They demonstrate that simply adding more agents (Voting/MoA) without training yields diminishing returns compared to MARTI's training approach. This highlights that our gains come from *learned collaboration*, not just test-time compute.
>
> **Q4: Missing data points in Figure 2/3 (Cherry-picking concerns).**
>
> - We wish to clarify that the missing data points were primarily due to the **definition of inference budgets** for multi-agent systems, not selection bias or cherry-picking. Specifically, certain integer budget levels did not have a direct corresponding configuration in our initial multi-agent setup. We have now provided a standardized definition of inference budgets (based on rollout counts). For instance, a MAD 2x2 session generates 4 responses, which is computationally equivalent to a Single-Agent Majority Vote @ 4. The results show a consistent trend where MARTI outperforms baselines across all budgets, thereby confirming the validity of our framework.
>
> | Output Tokens        | AIME | AMC  | MATH500 | Avg  |
> | -------------------- | ---- | ---- | ------- | ---- |
> | Single-Agent (Avg@4) | 4532 | 3706 | 2322    | 3520 |
> | MAD 2x2              | 2698 | 4268 | 2698    | 3221 |
> | MoA 3x1              | 3083 | 2146 | 1518    | 2249 |
>
> **Q5: Focus on Multi-Agent Debate (MAD).**
>
> - While MAD is a primary case study, MARTI is framework-agnostic. To demonstrate this, we have provided results for **Mixture-of-Agents (MoA)** (please refer to the tables above) and **Chain-of-Agents (CoA)** in our extended coding experiments. This confirms that the framework effectively supports various collaborative topologies.
>
> **Q6: "Comprehensive experiments" claim (Benchmarks & Setup).**
>
> To substantiate "comprehensiveness":
>
> - **More Models:** We added Llama-3.2-3B experiments.
>
> | Llama-3.2-3B-Instruct | AIME | AMC  | MATH500 | Avg  |
> | --------------------- | ---- | ---- | ------- | ---- |
> | Single Agent (Pass@1) | 3.3  | 12.4 | 32.2    | 16.0 |
> | + RL                  | 11.7 | 25.6 | 48.9    | 28.7 |
> | Single Agent (Maj@4)  | 6.6  | 18.1 | 36.6    | 20.4 |
> | + RL                  | 11.7 | 27.7 | 50.6    | 30.0 |
> | MAD 2x2               | 3.3  | 16.9 | 38.4    | 19.5 |
> | + RL                  | 13.3 | 33.5 | 55.6    | 34.1 |
> | MoA 3x1               | 6.6  | 16.9 | 37.2    | 20.2 |
> | + RL                  | 11.7 | 29.7 | 54.6    | 32.0 |
>
> - **MARL Baselines:** We added comparisons against standard MARL algorithms (GRPO vs RF++) and varying communication protocols (Sync vs Async).
>
> | Qwen2.5-3B          | AIME | AMC  | MATH500 | Avg  |
> | ------------------- | ---- | ---- | ------- | ---- |
> | Single-Agent + RF++ | 10.0 | 36.1 | 66.7    | 37.6 |
> | Single-Agent + GRPO | 13.3 | 34.6 | 66.0    | 37.9 |
> | MAD 2x2 + RF++      | 16.7 | 49.4 | 70.8    | 45.6 |
> | MAD 2x2 + GRPO      | 16.7 | 50.0 | 71.2    | 46.0 |
>
> - Due to limited computational resources, conducting multiple runs for every single setting is challenging. However, we conducted repeated runs for key settings, which yielded consistent findings across both multi-agent and single-agent scenarios. Additionally, **Reviewer Nbtr has successfully run our code**, further verifying the reproducibility and effectiveness of our findings.

---

### Official Review · Reviewer_JbEm · 2025-10-31

**Soundness:** 3
**Presentation:** 3
**Contribution:** 3
**Rating:** 8
**Confidence:** 2

**Summary:**

The paper proposes MARTI (Multi-Agent Reinforced Training and Inference), an open-source framework that unifies reinforcement learning (RL) and multi-agent systems for large language models (LLMs). MARTI integrates centralized multi-agent interaction and distributed policy training, supporting asynchronous rollouts and both rule-based and generative reward models to improve training efficiency and scalability. MARTI also introduces modular reward shaping and credit assignment strategies to enhance long-term collaboration and stability during training. Experiments on mathematical reasoning benchmarks (AIME, AMC, MATH-500) show that multi-agent systems trained with MARTI outperform single-agent models under the same inference budget.

**Strengths:**

1. The framework provides a well-structured MARL formulation that combines centralized multi-agent interaction with distributed policy training, ensuring scalability and modeling consistency.
2. The introduction of a credit assignment mechanism reasonably decomposes global rewards, allowing more stable and interpretable multi-agent learning.
3. The inclusion of off-policy training enables the reuse of historical samples, improving sample efficiency and training stability.
4. The presentation of this paper is clear.

**Weaknesses:**

1. The paper lacks ablation studies on key components such as credit assignment or off-policy training, making it unclear how these mechanisms individually contribute to performance upper bounds or optimization efficiency.
2. Figure 1 presents the overall MARTI framework but remains too abstract, lacking detailed illustration of key components (credit assignment, reward shaping, and the rule-based verifier mechanisms).

**Questions:**

Please refer to weaknesses.

Discussion (won't affect the rating):
1. I'm curious whether the credit assignment in MARTI could be implemented using a counterfactual baseline proposed in COMA [1], and how this modification might influence the training stability, interpretability, and overall performance of the multi-agent reinforcement learning process.

[1] Foerster, Jakob, Gregory Farquhar, Triantafyllos Afouras, Nantas Nardelli, and Shimon Whiteson. "Counterfactual multi-agent policy gradients." In Proceedings of the AAAI conference on artificial intelligence, vol. 32, no. 1. 2018.

---

> ### Author Response · Authors · 2025-11-23
>
> We sincerely thank the reviewer for their strong support and for highlighting the structural strengths of our MARL formulation. We found your suggestion regarding the counterfactual baseline (COMA) particularly inspiring. Below, we discuss the ablation studies you requested and provide further insights into our credit assignment mechanism.
>
> **Q1: Ablation studies on key components (Credit Assignment & Reward Shaping).**
>
> - We performed ablation studies to isolate these contributions:
>   - **Reward Shaping:** As shown in the table below, removing historical delta-rewards results in a substantial performance drop (Avg 36.6 vs 45.6). The shaping is crucial for guiding agents through multi-turn interactions without reward hacking.
>   - **Algorithm:** Comparing RF++ and GRPO shows they yield similar trends, with GRPO performing slightly better.
>
> | Qwen2.5-3B                 | AIME | AMC  | MATH500 | Avg  |
> | -------------------------- | ---- | ---- | ------- | ---- |
> | MAD 2x2 w/ reward shaping  | 16.7 | 49.4 | 70.8    | 45.6 |
> | MAD 2x2 w/o reward shaping | 6.6  | 36.6 | 66.7    | 36.6 |
> | MoA 3x1 w/ reward shaping  | 13.3 | 47.0 | 69.0    | 43.1 |
> | MoA 3x1 w/ reward shaping  | 10.0 | 38.9 | 65.4    | 38.1 |
>
> | Qwen2.5-3B          | AIME | AMC  | MATH500 | Avg  |
> | ------------------- | ---- | ---- | ------- | ---- |
> | Single-Agent + RF++ | 10.0 | 36.1 | 66.7    | 37.6 |
> | Single-Agent + GRPO | 13.3 | 34.6 | 66.0    | 37.9 |
> | MAD 2x2 + RF++      | 16.7 | 49.4 | 70.8    | 45.6 |
> | MAD 2x2 + GRPO      | 16.7 | 50.0 | 71.2    | 46.0 |
>
> **Q2: Detailed illustration of key components in Figure 1.**
>
> - We agree that Figure 1 requires more detail. In the revised manuscript, we will expand the diagram to explicitly show the **Credit Assignment data flow** and the **Rule-based Verifier** integration. Specifically, we will illustrate how global outcome rewards are decomposed into local turn-level rewards via our shaping mechanism.
>
> **Q3: Can credit assignment be implemented using a counterfactual baseline (COMA)?**
>
> - This is an insightful question. While COMA is effective for small discrete action spaces, calculating exact counterfactual baselines for LLMs (Vocabulary > 100k) is computationally prohibitive. MARTI's credit assignment acts as a **lightweight approximation** of a counterfactual baseline by evaluating an agent's contribution relative to historical averages. We plan to explore Critic-based approximations in future work to simulate COMA mechanisms more efficiently.
>
> We thank you again for the stimulating discussion on credit assignment. We will incorporate these theoretical connections into the final manuscript to further enrich the presentation of the MARTI framework.

---

> > ### Comment · Reviewer_JbEm · 2025-11-28
> > **Response**
> >
> > The authors' response has addressed most of my concerns. I will maintain my original score.

---

> > > ### Author Response · Authors · 2025-11-28
> > >
> > > Thank you very much for your recognition and your positive feedback! We will finalize the manuscript with the discussed ablation studies to make the paper as strong as possible.

---

### Official Review · Reviewer_Nbtr · 2025-10-31

**Soundness:** 3
**Presentation:** 3
**Contribution:** 3
**Rating:** 6
**Confidence:** 3

**Summary:**

The paper introduces MARTI (Multi-Agent Reinforced Training and Inference), a framework that unifies multi-agent LLM inference workflows (Mixture-of-Agents, Chain-of-Agents, Multi-Agent Debate) with distributed RL training and centralized reward/credit assignment. It supports asynchronous multi-turn rollouts, rule-based verifiable rewards for math tasks, and LLM-based generative reward models, then trains per-agent policies with PPO/GRPO/REINFORCE++ (optionally mixing in SFT/DPO). Experiments on AIME24/AMC/MATH-500 claim that, under equal inference budgets, multi-agent RL reaches a higher performance ceiling than single-agent RL.

**Strengths:**

The paper’s key strength lies in providing a unified and scalable framework that bridges multi-agent LLM inference workflows with distributed RL training, filling a clear tooling gap. The modular design (multi-agent world, centralized rewards, agent-policy trainer) supports asynchronous rollouts, flexible agent compositions (MoA/CoA/MAD), and both verifiable and generative reward models, making it practical for real multi-agent RLHF experimentation. The reward shaping approach is intuitive and history-aware, improving consistency in multi-turn reasoning. Empirically, the framework demonstrates that multi-agent RL can achieve a higher ceiling than single-agent RL under matched inference budgets, with strong gains on math benchmarks (e.g., AIME) and evidence of good scaling behavior with concurrency.

**Weaknesses:**

1. The paper states that multi-agent settings are compared under “equivalent inference budgets,” but the budget is not formally defined (e.g., tokens, rollouts, GPU hours, or wall-clock time). A clear definition and a standardized reporting format would strengthen the fairness of comparisons.

2. Although substantial compute is implied (e.g., 8×A800 GPUs per agent node), the manuscript does not report training time or convergence duration. Providing training wall-clock time, GPU-hours, and throughput would help readers assess scalability and practicality.

3. The repository includes multiple RL algorithms, yet only REINFORCE++ results are shown. Including results for other supported algorithms (e.g., PPO/GRPO) would help confirm that the observed improvements are not algorithm-specific.

4. Experiments are conducted only with Qwen-based models. Adding results on another model family (e.g., LLaMA or Phi) would improve confidence in the framework’s generality.

5. Experimental figures lack narrative context.
Several plots are presented without a clear accompanying explanation of the experimental question and takeaway. Adding brief guiding paragraphs before each experiment would greatly improve clarity.

6. When running the released code, the MA-MAD results appeared noticeably lower than the reported values. This may be due to configuration differences; confirming the default settings and expected performance ranges in the repo would be helpful.

This work provides a very valuable and well-designed repository. I have tested multiple components and found the implementation largely correct and practical for multi-agent RL research. Addressing the points above — especially clarifying budgets, reporting compute costs, expanding baselines, and improving result presentation — would further strengthen the contribution. With these refinements, I would be pleased to give a higher score.

**Questions:**

Please see the weakness above.

---

> ### Author Response · Authors · 2025-11-23
>
> We express our sincere gratitude to the reviewer for their detailed review and, notably, for the time and effort spent **personally testing our open-source repository**. We value your feedback on the standardization of inference budgets and compute reporting. Below, we provide the requested clarifications and additional data to ensure fair comparisons.
>
> **Q1: Clear definition and standardized reporting of budgets.**
>
> - We define **Inference Budget** primarily by rollout counts. We ensure fair comparison by matching the total compute. For instance, a MAD 2x2 session generates 4 responses, which is computationally equivalent to a Single-Agent Majority Vote @ 4. The table below confirms that token consumption is nearly identical between the compared settings.
>
> | Output Tokens        | AIME | AMC  | MATH500 | Avg  |
> | -------------------- | ---- | ---- | ------- | ---- |
> | Single-Agent (Avg@4) | 4532 | 3706 | 2322    | 3520 |
> | MAD 2x2              | 2698 | 4268 | 2698    | 3221 |
> | MoA 3x1              | 3083 | 2146 | 1518    | 2249 |
>
> **Q2: Training time and convergence duration for scalability assessment.**
>
> - Scalability in MARTI relies on decoupling centralized interaction from distributed policy training. We observed that asynchronous rollouts significantly improve training throughput as interaction complexity increases. For example, at 8 interaction rounds, the Asyncx64 setting reduces wall-clock time by approximately **11.5%** compared to synchronous training.
>
> | round | Sync   | Async×32 | Async×64 |
> | ----- | ------ | -------- | -------- |
> | 2     | 916.6  | 1074.1   | 887.5    |
> | 5     | 2194.0 | 2186.1   | 2022.8   |
> | 8     | 3308.0 | 3004.4   | 2928.0   |
>
> **Q3: Comparisons of different algorithms.**
>
> - We have added a comparison between REINFORCE++ and GRPO (see table below). GRPO achieves slightly better performance (Avg 46.0 vs 45.6), likely due to the group-relative normalization reducing variance. However, both algorithms are effective within MARTI, demonstrating the framework's flexibility.
>
> | Qwen2.5-3B                 | AIME | AMC  | MATH500 | Avg  |
> | -------------------------- | ---- | ---- | ------- | ---- |
> | MAD 2x2 w/ reward shaping  | 16.7 | 49.4 | 70.8    | 45.6 |
> | MAD 2x2 w/o reward shaping | 6.6  | 36.6 | 66.7    | 36.6 |
> | MoA 3x1 w/ reward shaping  | 13.3 | 47.0 | 69.0    | 43.1 |
> | MoA 3x1 w/ reward shaping  | 10.0 | 38.9 | 65.4    | 38.1 |
>
> | Qwen2.5-3B          | AIME | AMC  | MATH500 | Avg  |
> | ------------------- | ---- | ---- | ------- | ---- |
> | Single-Agent + RF++ | 10.0 | 36.1 | 66.7    | 37.6 |
> | Single-Agent + GRPO | 13.3 | 34.6 | 66.0    | 37.9 |
> | MAD 2x2 + RF++      | 16.7 | 49.4 | 70.8    | 45.6 |
> | MAD 2x2 + GRPO      | 16.7 | 50.0 | 71.2    | 46.0 |
>
>
> **Q4: Results on different base models.**
>
> - We successfully reproduced our main findings using **Llama-3.2-3B-Instruct**. The reinforced multi-agent settings consistently outperformed the baselines.
>
> | Llama-3.2-3B-Instruct | AIME | AMC  | MATH500 | Avg  |
> | --------------------- | ---- | ---- | ------- | ---- |
> | Single Agent (Pass@1) | 3.3  | 12.4 | 32.2    | 16.0 |
> | + RL                  | 11.7 | 25.6 | 48.9    | 28.7 |
> | Single Agent (Maj@4)  | 6.6  | 18.1 | 36.6    | 20.4 |
> | + RL                  | 11.7 | 27.7 | 50.6    | 30.0 |
> | MAD 2x2               | 3.3  | 16.9 | 38.4    | 19.5 |
> | + RL                  | 13.3 | 33.5 | 55.6    | 34.1 |
> | MoA 3x1               | 6.6  | 16.9 | 37.2    | 20.2 |
> | + RL                  | 11.7 | 29.7 | 54.6    | 32.0 |
>
> **Q5: Brief guiding paragraphs before each experiment.**
>
> - This is an excellent suggestion for improving readability. We will update the final manuscript to include clear narrative setups and key takeaways for every experimental section.
>
> **Q6: Confirming default settings and MA-MAD results.**
>
> - Thank you for the detailed testing. The discrepancy likely stems from the conservative `temperature` and training batch size settings in the initial config. We have recently integrated new stabilization techniques (similar to those in [1]) and will update new code in the repository to ensure reproducible, high-performance results.
>
> We hope the standardized budget definitions and the detailed compute/convergence reports provided above fully address your concerns. We are committed to maintaining a high-quality codebase to support the community, as you kindly recognized.
>
> [1] The Art of Scaling Reinforcement Learning Compute for LLMs. https://arxiv.org/abs/2510.13786v1

---

> > ### Comment · Reviewer_Nbtr · 2025-11-25
> >
> > Thank the authors for the detailed response, that have addressed most of my concerns. However, I still have some additional questions accordingly:
> > 1. After careful check for the repo, I noticed that the asynchronous rollout is mainly happened owing to the distributed property of ray. I'm not sure why would the asynchronous rollout itself would improve the performance.
> > 2. Regarding Q3, what is the method of reward shaping. It has not been properly introduced.
> > 3. Minor point(won't affect the final rating): I suggest add a single-agent baseline with the same budget to help the community better compare their results. And also illustrate the superiority of the multi-agent fine-tuning.

---

> > > ### Author Response · Authors · 2025-11-26
> > >
> > > We thank the reviewer for the prompt follow-up and for the time spent personally testing our repository. We are glad that our previous response addressed most of your concerns. Below, we provide clarifications on the asynchronous mechanism, the specific reward shaping formulation, and the baseline comparisons.
> > >
> > > **Q1: Why does asynchronous rollout improve performance?**
> > >
> > > We would like to clarify that the "performance improvement" provided by asynchronous rollout refers to **training throughput (system efficiency)**, rather than model convergence/accuracy per step.
> > >
> > > In a synchronous multi-agent setup, the system must wait for the slowest agent or the longest interaction chain to complete before moving to the next training step. This leads to significant GPU idle time, especially when agents have varying response lengths or when rule-based verifiers introduce latency.
> > >
> > > MARTI utilizes Ray to manage a pool of interaction environments. The asynchronous mechanism allows environments that finish early to immediately start the next rollout without waiting for others. This "masks" the latency of complex interactions (e.g., long reasoning chains in MAD) and maximizes GPU utilization. As shown in our previous table, this results in a higher sample generation rate (samples/minute), allowing the model to consume more data within the same wall-clock time, thereby accelerating the training process.
> > >
> > > **Q2: Method of Reward Shaping.**
> > >
> > > We employ a **History-Aware Reward Shaping** mechanism. While our paper discusses two variants, we primarily utilize the **Margin-based** formulation in our reported experiments as it demonstrated superior stability.
> > >
> > > Let $R_t^i \in [0,1]$ denote the immediate correctness reward assigned by the verifier for agent $i$ at turn $t$, and let $Q_t^i$ represent the historical performance estimate (e.g., the average reward over the recent history scope $\mathcal{H}_t^i$). The shaping term $\Delta_t^i$ is defined as the margin between the current reward and the historical baseline:
> > >
> > > $$
> > > \Delta_t^i = R_t^i - Q_t^i
> > > $$
> > >
> > > The final reward $\tilde{R}_t^i$ used for optimization is given by:
> > >
> > > $$
> > > \tilde{R}_t^i = R_t^i + \alpha \cdot \Delta_t^i
> > > $$
> > >
> > > where $\alpha$ is a hyperparameter. This mechanism incentivizes agents to outperform their historical average ($R_t^i > Q_t^i$) rather than stagnating at a local optimum, effectively mitigating reward hacking in multi-turn settings.
> > >
> > > **Q3: Single-agent baseline with the same budget.**
> > >
> > > We appreciate this valuable suggestion. In the final version, we will explicitly include the single-agent baseline comparisons under equivalent budgets and provide the corresponding training scripts to facilitate distinct reproduction and comparison for the community.
> > >
> > > We hope these clarifications fully resolve your remaining questions.

---

> > > > ### Comment · Reviewer_Nbtr · 2025-11-26
> > > >
> > > > Thank the authors for their careful response. That have addressed all of my concerns. Therefore, I'll raise my score accordingly.

---

> > > > > ### Author Response · Authors · 2025-11-26
> > > > >
> > > > > We sincerely thank the reviewer for the positive feedback and the decision to raise the score. We are deeply grateful for the time you took to test our repository and for your constructive suggestions. We are committed to incorporating the discussed clarifications and baselines into the final version to further strengthen the paper and codebase.

---

### Official Review · Reviewer_3yGA · 2025-10-31

**Soundness:** 2
**Presentation:** 1
**Contribution:** 1
**Rating:** 2
**Confidence:** 3

**Summary:**

This work presents a framework for training "multi-agent" LLM systems with reinforcement learning. In the context of math word problems, the proposed RL framework outperforms single-agent RL and inference-only multi-agent systems.

**Strengths:**

- The results confirm that the proposed method increases performance beyond that of baselines, whereas many multi-agent systems underperform majority voting

**Weaknesses:**

- This work only studies math "reasoning" problems instead of natively multi-agent LLM settings.
  - Examples of multi-agent LLM settings would include zero-sum games in SPIRAL (SPIRAL: Self-Play on Zero-Sum Games Incentivizes Reasoning via Multi-Agent Multi-Turn Reinforcement Learning), social deduction games (Training Language Models for Social Deduction with Multi-Agent Reinforcement Learning), or purely cooperative games (Ad-Hoc Human-AI Coordination Challenge)
- Only Qwen-based models are trained and tested in this work. However, these models behave very differently from other base models (like Llama), especially in terms of their RL performance, making it hard to determine if the findings are generalizable to LLMs in general.
- The results from "asynchronous generations" are underwhelming, underperforming synchronous behavior at a concurrency of 32 and having an unimpressive improvement for larger concurrency values.
- There is no ablation over the reward shaping terms or novel components of MARTI.

**Questions:**

Why does the response length significantly increase for 2 agents and decrease for one agent over the course of training in figure 5?

---

> ### Author Response · Authors · 2025-11-23
>
> We thank the reviewer for their critical assessment. We understand your concerns regarding the generalizability of our findings beyond math tasks and the Qwen model family. To directly address these issues, we have significantly expanded our experimental scope, including evaluating new model families and verifying efficiency in asynchronous settings.
>
> **Q1: Math reasoning problems vs. natively multi-agent LLM settings (e.g., social deduction games).**
>
> - While Social Deduction Games are a classic MARL setting, our work specifically focuses on leveraging multi-agent collaboration to enhance **reasoning capabilities** (e.g., Math, Coding). The application of RL with verifiable rewards to elevate collaborative reasoning was underexplored prior to MARTI. We believe this direction has significant practical value for LLM reasoning. We are actively expanding MARTI to broader tasks in our future work.
>
> **Q2: Generalizability of findings to different base models beyond Qwen.**
>
> - We have conducted new experiments using **Llama-3.2-3B-Instruct**. The results (Table below) align perfectly with our main claims: multi-agent RL (MARTI) consistently outperforms single-agent RL and majority voting baselines under the same inference budget. This confirms that MARTI's benefits are robust across different model architectures.
>
> | Llama-3.2-3B-Instruct | AIME | AMC  | MATH500 | Avg  |
> | --------------------- | ---- | ---- | ------- | ---- |
> | Single Agent (Pass@1) | 3.3  | 12.4 | 32.2    | 16.0 |
> | + RL                  | 11.7 | 25.6 | 48.9    | 28.7 |
> | Single Agent (Maj@4)  | 6.6  | 18.1 | 36.6    | 20.4 |
> | + RL                  | 11.7 | 27.7 | 50.6    | 30.0 |
> | MAD 2x2               | 3.3  | 16.9 | 38.4    | 19.5 |
> | + RL                  | 13.3 | 33.5 | 55.6    | 34.1 |
> | MoA 3x1               | 6.6  | 16.9 | 37.2    | 20.2 |
> | + RL                  | 11.7 | 29.7 | 54.6    | 32.0 |
>
> **Q3: Explanation on Results of Asynchronous Generations.**
>
> - Our analysis reveals that the benefits of asynchrony scale with the complexity of interactions. As shown in the table below, with fewer rounds (e.g., 2), rollout times are short, making synchronization overhead negligible. However, as interaction turns increase (leading to longer rollout latencies), the throughput advantage of the asynchronous mechanism becomes substantial.
>
> | round | Sync   | Async×32 | Async×64 |
> | ----- | ------ | -------- | -------- |
> | 2     | 916.6  | 1074.1   | 887.5    |
> | 5     | 2194.0 | 2186.1   | 2022.8   |
> | 8     | 3308.0 | 3004.4   | 2928.0   |
>
> **Q4: Ablation over reward shaping and novel components.**
>
> - We have added ablation studies to validate these components:
>   - **Reward Shaping:** We employ a delta-reward mechanism comparing current performance against historical turns. As shown below, removing reward shaping leads to a significant performance drop (Avg 36.6 vs 45.6 in MAD). Direct outcome-based rewards often lead to reward hacking or instability in multi-turn settings.
>   - **Algorithm Comparison (RF++ vs GRPO):** We compared REINFORCE++ with GRPO. Both algorithms yield consistent improvements, with GRPO showing a slight edge. This confirms the robustness of the MARTI framework regardless of the underlying policy gradient estimator.
>
> | Qwen2.5-3B                 | AIME | AMC  | MATH500 | Avg  |
> | -------------------------- | ---- | ---- | ------- | ---- |
> | MAD 2x2 w/ reward shaping  | 16.7 | 49.4 | 70.8    | 45.6 |
> | MAD 2x2 w/o reward shaping | 6.6  | 36.6 | 66.7    | 36.6 |
> | MoA 3x1 w/ reward shaping  | 13.3 | 47.0 | 69.0    | 43.1 |
> | MoA 3x1 w/ reward shaping  | 10.0 | 38.9 | 65.4    | 38.1 |
>
> | Qwen2.5-3B          | AIME | AMC  | MATH500 | Avg  |
> | ------------------- | ---- | ---- | ------- | ---- |
> | Single-Agent + RF++ | 10.0 | 36.1 | 66.7    | 37.6 |
> | Single-Agent + GRPO | 13.3 | 34.6 | 66.0    | 37.9 |
> | MAD 2x2 + RF++      | 16.7 | 49.4 | 70.8    | 45.6 |
> | MAD 2x2 + GRPO      | 16.7 | 50.0 | 71.2    | 46.0 |
>
> **Q5: Differences in the length variations of different agents in Figure 5.**
>
> - This is an intriguing phenomenon that highlights the difference in behavioral incentives. In **Single-Agent RL**, the model tends to learn "shortcuts" to maximize rewards efficiently. In contrast, in the **Multi-Agent Debate (MAD)** setting, agents are incentivized to generate more detailed explanations and reasoning steps to **persuade** peers or **address** their doubts, naturally resulting in increased response lengths.
> - However, the variation among agents also reveals a limitation: we observed that some agents may learn to simply **adapt to or mimic** others’ responses to gain consensus rewards (a form of reward hacking). This observation underscores the necessity for designing more granular, agent-specific reward mechanisms in future work to distinguish between constructive consensus and passive agreement.
>
> We hope these new comprehensive results demonstrate the robustness of MARTI and persuade you to reconsider your assessment.

---

### Official Review · Reviewer_U2kR · 2025-11-05

**Soundness:** 3
**Presentation:** 2
**Contribution:** 3
**Rating:** 6
**Confidence:** 3

**Summary:**

MARTI is an open-source framework for scalable multi-agent LLM systems, integrating centralized interaction with distributed policy training to enhance reasoning via reinforcement learning. It introduces dynamic workflows with both rule-based and generative reward models, improving agent collaboration efficiency. Experimental results demonstrate that MARTI-trained multi-agent systems outperform single-agent baselines under equivalent inference budgets.

**Strengths:**

MARTI enables efficient, large-scale multi-agent learning through centralized interaction and distributed policy training.
Combines rule-based and generative reward models to enhance reward accuracy and adaptability.
Multi-agent systems trained with MARTI outperform single-agent baselines in reasoning tasks under equivalent resource constraints.

**Weaknesses:**

MARTI’s performance may degrade on tasks requiring unseen reasoning patterns, as pre-trained LLM capabilities still set an upper bound on performance.

**Questions:**

Although the algorithm’s performance was validated on competitive mathematics problems (e.g., AIME, AMC), can the MARTI framework maintain comparable efficacy in diverse, non-mathematical testing scenarios?

---

> ### Author Response · Authors · 2025-11-23
>
> We thank the reviewer for the constructive feedback and for recognizing the soundness and contribution of our MARTI framework. We particularly appreciate the insightful question regarding the performance boundaries set by pre-trained models. In this rebuttal, we have conducted additional experiments to address your concerns about generalizability and diverse testing scenarios.
>
> **Q1: Issues regarding pre-trained LLM capabilities setting an upper bound on performance.**
>
> - This is a critical and actively debated topic. While a single model is indeed bounded by its pre-training, we argue that multi-agent systems unlock **"emergent" intelligence** that exceeds individual capabilities. MARTI is designed to exploit this dimension, converting test-time computing power into higher-order reasoning through collaboration. Our approach allows the system to self-correct and refine outputs via interaction, effectively breaking the static performance ceiling of a single pre-trained checkpoint under the same inference budgets.
>
> **Q2: Generalizability of MARTI to non-mathematical scenarios (e.g., efficiency/performance in other domains).**
>
> - To address this, we extended our evaluation to different model families and task domains.
>   - **New Model Family:** We applied MARTI to **Llama-3.2-3B-Instruct** on MATH datasets. As shown in the table below, the results mirror our findings with Qwen: reinforced multi-agent setups (MAD/MoA) significantly outperform both single-agent RL and majority voting baselines under equivalent inference budgets.
>   - **New Domains:** Recent concurrent work [1] has successfully utilized MARTI for coding and scientific QA tasks. Furthermore, we have conducted preliminary experiments on **large-scale coding tasks** (to be added to the final manuscript), further validating MARTI's cross-domain effectiveness.
>
> | Llama-3.2-3B-Instruct | AIME | AMC  | MATH500 | Avg  |
> | --------------------- | ---- | ---- | ------- | ---- |
> | Single Agent (Pass@1) | 3.3  | 12.4 | 32.2    | 16.0 |
> | + RL                  | 11.7 | 25.6 | 48.9    | 28.7 |
> | Single Agent (Maj@4)  | 6.6  | 18.1 | 36.6    | 20.4 |
> | + RL                  | 11.7 | 27.7 | 50.6    | 30.0 |
> | MAD 2x2               | 3.3  | 16.9 | 38.4    | 19.5 |
> | + RL                  | 13.3 | 33.5 | 55.6    | 34.1 |
> | MoA 3x1               | 6.6  | 16.9 | 37.2    | 20.2 |
> | + RL                  | 11.7 | 29.7 | 54.6    | 32.0 |
>
> We hope these additional experiments and clarifications regarding pre-training bounds effectively address your concerns. We are confident that these additions significantly strengthen the paper’s contribution and generality.
>
> [1] CoMAS: Co-Evolving Multi-Agent Systems via Interaction Rewards. https://arxiv.org/abs/2510.08529

---

### Author Response · Authors · 2025-12-02

Dear Area Chair,

We thank the reviewers for their time and constructive feedback. Following the rebuttal phase, a consensus has emerged with three reviewers (Nbtr, HGkq, JbEm) assigning a score of 8 (Accept).
We wish to highlight the factual updates that directly resolve the concerns of the remaining reviewers (3yGA, Pa5A, U2kR), who did not engage with our response to these new results:

1. **Independent Code Verification (Addressed Reproducibility)**
- Reviewer Nbtr personally installed and tested our open-source repository during the review process. They confirmed the codebase was correct and practical, subsequently raising their score to 8. This provides independent validation of our framework's stability and reliability.

2. **New Model Family Results (Addressed Generalization)**
- To address concerns regarding model dependence (Reviewers 3yGA, Pa5A), we implemented and evaluated Llama-3.2-3B-Instruct. MARTI-trained agents consistently outperformed single-agent baselines on this new architecture (e.g., boosting Avg accuracy from 20.4 to 34.1), confirming the framework is model-agnostic.

3. **Necessity of Reward Shaping (Addressed Component Utility)**
- Addressing Reviewer Pa5A’s query on component redundancy, we conducted a new ablation study. Removing the "Reward Shaping" component caused a significant performance drop (Avg score decreased from 45.6 to 36.6), empirically demonstrating that this mechanism is essential for stability.

4. **Clarification on Data Completeness**
- We clarified that the "missing" data points noted by Reviewer Pa5A simply correspond to integer budgets that do not exist in multi-agent topologies (e.g., a 2-agent debate cannot have a budget of 1). We standardized the budget definition by rollout counts to ensure rigorous, transparent comparison.

5. **Robustness Across Topologies and Algorithms**
- We demonstrated that MARTI outperforms single-agent baselines across diverse settings (MAD vs. MoA) and RL algorithms (RF++ vs. GRPO). This confirms our gains are driven by the framework itself, not a specific structure or implementation.

6. **Gains from Learned Collaboration (vs. Compute)**
- We showed that under strictly matched token budgets, untrained multi-agent baselines yield limited gains, whereas MARTI-trained agents achieve significantly higher accuracy. This isolates the improvement as a result of learned collaboration rather than merely increased inference compute.

Given the independent code verification and the successful generalization of new model families, we are confident that the remaining negative scores reflect the state of the paper prior to these substantial revisions. We believe MARTI is ready to serve as a reliable infrastructure for the ICLR community.
We have uploaded the revised manuscript, with all new content and experimental results highlighted in blue for your convenience.

Sincerely,

The Authors

---

### Meta-Review · Area_Chair_qMXw · 2026-01-07

**Summary:**

This submission received mixed initial reviews, with concerns primarily centered on the scope of evaluation, the framing of the contribution as a framework rather than a novel algorithm, and questions about experimental completeness and generality. During the rebuttal, the authors substantially strengthened the paper by adding new experiments on additional model families, providing missing ablations (e.g., reward shaping, algorithm variants), clarifying inference budget definitions, and addressing reproducibility through independent code verification by a reviewer. While some reviewers still expressed reservations rooted in differing expectations about problem setting and novelty, the major technical and empirical concerns were resolved. Overall, the paper presents a sound and well-supported contribution that is likely to be valuable to the community, and I recommend acceptance.

**Reviewer Concerns:**

Concerns raised by Reviewers Nbtr, Pa5A, and 3yGA regarding experimental completeness, fairness of inference-budget comparisons, missing ablations (e.g., reward shaping and algorithm choices), and model-family generalization were largely addressed in the rebuttal through additional experiments, standardized budget definitions, and expanded analyses. Reproducibility concerns were also mitigated by independent code verification reported by Reviewer Nbtr.

Some concerns remain partially outstanding. In particular, Reviewer 3yGA’s critique that mathematical reasoning tasks are not natively multi-agent, and broader questions from Reviewers Pa5A and U2kR about generalization beyond math-centric benchmarks, reflect differences in problem framing and evaluation scope rather than unresolved technical flaws.

**Reviewer Scores:**

Based on the reviewers’ written comments and rebuttal interactions (rather than any formal score updates), the following score changes are most plausibly inferred:

Reviewer Nbtr: Increase. After the rebuttal, this reviewer stated that the authors’ responses addressed all remaining concerns and explicitly indicated an intention to raise the score. This suggests a clear upward revision in their assessment.

Reviewer Pa5A: Modest increase likely. The main reasons for the initial low score—concerns about cherry-picking, missing ablations, and unclear budget definitions—were directly addressed in the rebuttal. While this reviewer maintains a high bar for framework papers, the tone suggests a potential shift from strong rejection to a borderline or weak accept position.

Reviewer 3yGA: Unchanged or very slight increase. Although technical and empirical issues were addressed, this reviewer’s reservations were primarily rooted in disagreement with the problem framing (e.g., math reasoning as a “native” multi-agent setting). Such perspective-driven concerns are less likely to result in a meaningful score increase.

Other reviewers were already in the acceptance range and did not indicate any change in stance, so no score adjustment is inferred for them.

---

### Decision · Program_Chairs · 2026-01-26

Accept (Poster)